# The transferability of adjoint inversion products between different ice flow models

Jowan M. Barnes[1], Thiago Dias dos Santos[2,3], Daniel Goldberg[4], G. Hilmar Gudmundsson[1], Mathieu Morlighem[2], and Jan De Rydt[1]

[1]Department of Geography and Environmental Sciences, Northumbria University, Newcastle upon Tyne, UK
[2]Department of Earth System Science, University of California, Irvine, CA, USA
[3]Centro Polar e Climático, Universidade Federal do Rio Grande do Sul, Porto Alegre, RS, Brazil
[4]School of GeoSciences, University of Edinburgh, Edinburgh, UK

**Correspondence:** Jowan M. Barnes (jowan.barnes@northumbria.ac.uk)

**Abstract.** Among the most important challenges faced by ice flow models is how to represent basal and rheological conditions, which are challenging to obtain from direct observations. A common practice is to use numerical inversions to calculate estimates for the unknown properties, but there are many possible methods and not one standardised approach. As such, every ice flow model has a unique initialisation procedure. Here we compare the outputs of inversions from three different ice flow models, each employing a variant of adjoint-based optimisation to calculate basal sliding coefficients and flow rate factors using the same observed surface velocities and ice thickness distribution. The region we focus on is the Amundsen Sea Embayment in West Antarctica, the subject of much investigation due to rapid changes in the area over recent decades. We find that our inversions produce similar distributions of basal sliding across all models, despite using different techniques, implying that the methods used are highly robust and represent the physical equations without much influence by individual model behaviours. Transferring the products of inversions between models results in time-dependent simulations displaying variability on the order of or lower than existing model intercomparisons. Focusing on contributions to sea level, the highest variability we find in simulations run in the same model with different inversion products is 32%, over a 40 year period, a difference of $3.67\,\mathrm{mm}$. There is potential for this to be improved with further standardisation of modelling processes, and the lowest variability within a single model is 13%, or $1.82\,\mathrm{mm}$ over 40 years. While the successful transfer of inversion outputs from one model to another requires some extra effort and technical knowledge of the particular models involved, it is certainly possible and could indeed be useful for future intercomparison projects.

## 1 Introduction

Many ice flow models use inverse methods to calculate initial conditions for properties of the ice for which directly observed data do not exist, or are of poor quality. Inversion is an iterative process which starts from an initial guess and obtains improved

values for the unknown property based on its relationship to a well-observed property, such as surface velocity. This process is generally undertaken for at least one of ice rheology (flow rate factor, $A$), basal sliding and bed topography. The use of such methods in glaciology dates back to MacAyeal (1992), who used control methods to derive a distribution of basal friction under a tributary of Ross Ice Shelf, Ice Stream E (now known as MacAyeal Ice Stream). Since then, the use of inverse methods in estimating basal and internal conditions of glaciers from measured surface velocities has become widespread, supported by an increase in observational data from satellites and improvements in computational efficiency (Pattyn et al., 2017). The ability to perform large-scale inversions has revolutionised the field of ice flow modelling, allowing better representation of basal and rheological conditions to which the flow is sensitive. Several methods have been proposed and tested for models of varying complexity, including the adjoint method (MacAyeal, 1993) and subsequent variations (e.g., Vieli and Payne, 2003; Joughin et al., 2004; Petra et al., 2012; Morlighem et al., 2013; Perego et al., 2014), a least-squares inversion (Thorsteinsson et al., 2003), a non-linear Bayesian method (Raymond and Gudmundsson, 2009), inverse Robin problems (Arthern and Gudmundsson, 2010), a nudging method (Mosbeux et al., 2016) and an ensemble Kalman filter method (Gillet-Chaulet, 2020).

However, these inverse problems are not well-posed and a unique solution is never guaranteed, regardless of the method used. In fact, a given inverse problem may have an infinite number of different solutions producing identical outputs of the forward model (e.g., Zhdanov, 2015). An approach often used to remedy the ill-posedness of inverse problems is the introduction of regularisation, but there are many possible techniques for doing so. As such, the methods used and results obtained from inversions could differ considerably between models.

Aspects of inversion processes within individual models have been the subject of several recent studies. Koziol and Arnold (2017) incorporated subglacial hydrology into inversions for basal sliding. Kyrke-Smith et al. (2018) analysed the effects of basal topography on inversions. The sensitivity of inversions to several ice properties was tested by Zhao et al. (2018), and sensitivities at the surface to perturbations in basal conditions from inversions have been investigated by Martin and Monnier (2014) and Cheng and Lötstedt (2020). However, there are not many direct comparisons between inversions from different models. Morlighem et al. (2010) compared inversions using ice flow equations of varying complexity, and the initMIP-Antarctica exercise (Seroussi et al., 2019), as part of the ISMIP6 model intercomparison project, compared models which were set up using different datasets, with a focus on the responses in forward model runs to a variety of initialisation procedures.

The differences between inversion outputs from different modelling platforms have not been given attention under controlled conditions, as it is generally thought that the products of inversions are highly model-dependent. In model intercomparison projects (e.g., Bindschadler et al., 2013; Asay-Davis et al., 2016; Cornford et al., 2020) boundary conditions such as topography and melt rates are specified in detail, but participants are not given set values for the basal sliding coefficient or ice rheology rate factor. Instead, participants are asked to tune the initialisation of their models individually to set these values. This implies that the results of inversions are believed not to be purely representative of the physical properties of an ice flow, but also to account for non-physical model behaviours resulting from different numerical implementations or approximations.

We seek to test this belief, by comparing the outputs of carefully controlled inversions in different models.

For this study, the focus is on inversions for basal sliding coefficients and ice rheology rate factors using an adjoint method, and using the same input datasets. We compare the results of inversions from three ice flow models, identify the factors which cause differences between them and investigate the effect these differences have when transferring the products of inversions between models. We are interested in the extent to which the inversion processes are reflective of the physical ice flow described by the model equations, and by how much numerical model behaviour might be influencing the outputs. If the inversion outputs from the models are similar, we can be sure that they represent a solution to the given physical equations, without the results being heavily influenced by model-specific differences in the processes used. As part of our investigation, we will assess whether the products of inversions can be used outside their model of origin, and whether the fields produced by inversions from different models result in similar behaviour in transient simulations.

Our chosen study area is the Amundsen Sea Embayment (ASE) in West Antarctica (Fig. 1). Within this region, Thwaites Glacier is the subject of a targeted multidisciplinary investigation, the International Thwaites Glacier Collaboration (Scambos et al., 2017). Understanding change in the West Antarctic Ice Sheet has been identified as a top priority for future Antarctic research (National Academies of Sciences, Engineering, and Medicine, 2015). The Amundsen Sea, and Thwaites Glacier in particular, are of considerable interest due to rapid changes observed in the area over recent years (e.g., Mouginot et al., 2014; Milillo et al., 2019). Mass loss in the ASE is happening at a greater rate than anywhere else in Antarctica (Shepherd et al., 2018; Rignot et al., 2019), and has been accelerating over recent decades (Sutterley et al., 2014). Many model simulations have been used to make predictions of the future evolution of Thwaites Glacier and the ASE region, and they produce different results depending on model setup (e.g., Favier et al., 2014; Yu et al., 2018). However, these differences are predominantly in the rates of change rather than the direction of evolution. Forward simulations of ice flow models have been proved to be robust in intercomparison experiments, most recently MISMIP+ (Cornford et al., 2020), and they generally agree that the trend of rapid retreat in the ASE will continue into the future (e.g., Joughin et al., 2014; DeConto and Pollard, 2016). There is a constant effort to improve the understanding and functionality of all aspects of ice flow models, and to reduce uncertainty in their predictions. Among the most important factors which models must account for, and which are challenging to obtain from direct observation, are ice rheology and basal conditions.

In this work, we start by giving details of the models used and their respective inversion procedures in Sect. 2. We outline the experiments, along with the datasets and boundary conditions used, in Sect. 3. Following this, output fields of speed misfit, rate factor and the basal sliding coefficient from inversion runs in the three models are compared in Sect. 4. In order to better understand how individual model behaviours affect inversion results, we then investigate specific factors which cause the differences. Finally, in Sect. 5, we run transient simulations in each model using all three sets of inversion outputs, to assess the feasibility of transferring products of inversions between models and identify problems which may be encountered in doing so.

## 2 Model details

Three models are used in this study; Úa (Gudmundsson, 2020), the Ice-sheet and Sea-level System Model (Larour et al., 2012), known as ISSM, and the STREAMICE module of MITgcm (Goldberg and Heimbach, 2013). Úa and ISSM implement the Shallow Shelf Approximation (SSA) of MacAyeal (1989), and STREAMICE uses the L1L2 variant described in Goldberg (2011), in which some depth-dependency is retained in vertical shear terms and basal stress. Úa and ISSM are finite element models which employ unstructured meshes. These can be adapted to target specific areas of interest with finer resolution. STREAMICE, which inherits its grid and parallel domain decomposition from the MITgcm ocean model (Marshall et al., 1997), operates on a structured rectangular grid. It is a finite element code which uses bilinear basis functions to solve the

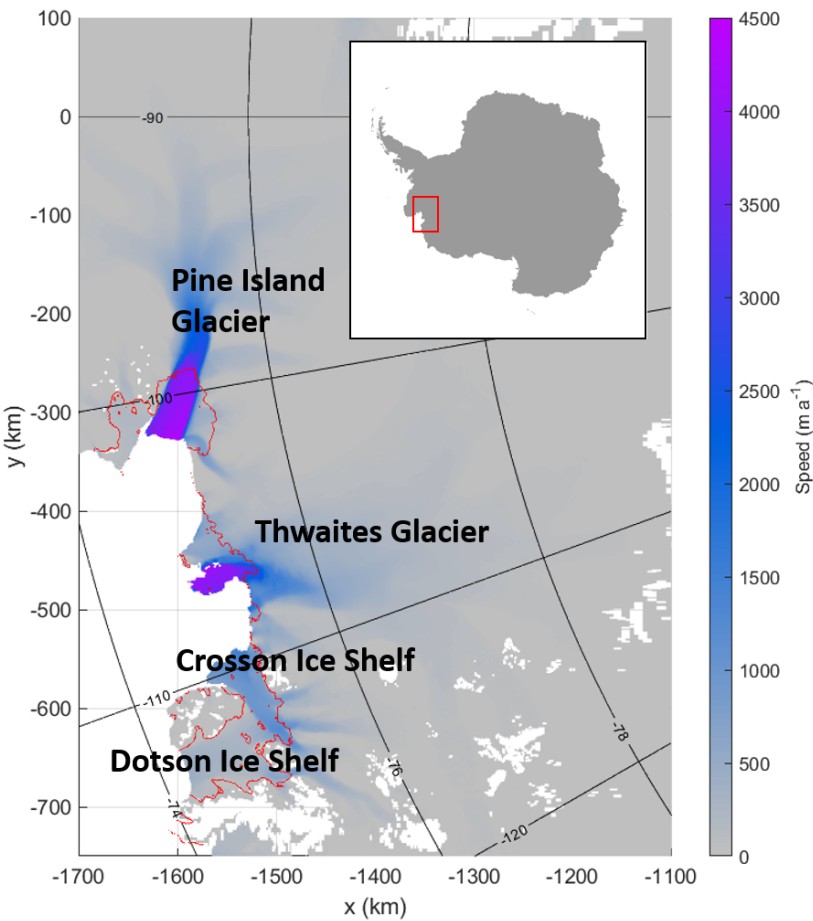

**Figure 1.** Amundsen Sea Embayment shaded with speed measurements from Mouginot et al. (2014). Grounding lines are shown in red.

momentum balance on its rectangular grid, and a zero-order discontinuous galerkin method to solve the mass balance. More details can be found in Goldberg and Heimbach (2013).

## 2.1 Parameters for inversion

Each model performs inversions for two parameters, a rheological parameter and a basal sliding coefficient. To describe ice rheology all models use the constitutive equation

$$\boldsymbol{\tau} = \frac{1}{2} A^{-\frac{1}{n}} \dot{\varepsilon}_e^{\frac{1-n}{n}} \dot{\boldsymbol{\varepsilon}} \tag{1}$$

generally referred to in glaciology as the Glen's flow law (Glen, 1958), where $\dot{\varepsilon}$ is the strain rate, $\dot{\varepsilon}_e$ is the effective strain rate, $A$ is the ice flow rate factor, $\boldsymbol{\tau}$ is the deviatoric stress and $n$ is a stress exponent. All the inversions in this study use the standard value of $n = 3$. Úa inverts for $A$ or $\log_{10} A$, while ISSM inverts for a rheological parameter $B = A^{-\frac{1}{n}}$, sometimes known as the associated rate factor (eg., Greve and Blatter, 2009), and STREAMICE inverts for $\sqrt{B}$. The rate factor is an indicator of how soft or damaged ice is, with higher values of $A$ corresponding to softer ice, and higher values of $B$ corresponding to stiffer ice.

All three models employ the Weertman sliding law (Weertman, 1957), albeit in a slightly different form, as follows:

$$\text{Úa}: \quad \boldsymbol{\tau_b} = (C + C_0)^{-\frac{1}{m}} (\|\boldsymbol{v_b}\|^2 + v_0^2)^{\frac{1-m}{2m}} \boldsymbol{v_b} \tag{2}$$

$$\text{ISSM}: \quad \boldsymbol{\tau_b} = \beta^2 \|\boldsymbol{v_b}\|^{\frac{1}{m}-1} \boldsymbol{v_b} \tag{3}$$

$$\text{STREAMICE}: \quad \boldsymbol{\tau_b} = \beta^2 (\|\boldsymbol{v_b}\|^2 + v_0^2)^{\frac{1-m}{2m}} \boldsymbol{v_b}, \tag{4}$$

where $\boldsymbol{\tau_b}$ is the basal stress, $\boldsymbol{v_b}$ is the basal velocity, $C_0$ and $v_0$ are regularisation constants and $m$ is the sliding law exponent, which in the case of these inversions is always $m = 3$. Úa inverts for either the basal sliding coefficient $C$, or $\log_{10} C$, while ISSM inverts for $\beta^2$, sometimes referred to as a basal friction or roughness coefficient, and STREAMICE inverts for $\beta$. In Úa, $C_0 = 1 \times 10^{-20} \, \text{kPa}^{-3} \, \text{m} \, \text{a}^{-1}$ and $v_0 = 1 \times 10^{-4} \, \text{m} \, \text{a}^{-1}$, and in STREAMICE, $v_0 = 1 \times 10^{-6} \, \text{m} \, \text{a}^{-1}$. ISSM does not employ a regularisation term in the sliding law, but the code contains a numerical verification which prevents division by zero.

## 2.2 Inversion methods

All of the inversion methods involve minimising a cost function of general form

$$\mathcal{J} = I + R, \tag{5}$$

where $I$ is a misfit function and $R$ is a regularisation term. The exact forms that these take varies. In the following, $\boldsymbol{p} = (p_1, p_2)$ refers to the parameters being inverted for, which differs between models. The observed values of surface velocities are $u_{\text{obs}}$ and $v_{\text{obs}}$, in the $x$- and $y$-directions on a polar stereographic grid, with observational errors $u_{\text{err}}$ and $v_{\text{err}}$.

All of the inversion methods contain regularisation parameters which must be chosen. L-curve analyses were performed for each model, and further details of these can be found in Appendix A.

### 2.2.1 Úa

In Úa, the cost function is $\mathcal{J}_{\text{Úa}} = I_{\text{Úa}} + R_{\text{Úa}}$. The misfit function is given by

$$I_{\text{Úa}} = \frac{1}{2\mathcal{A}} \int ((u - u_{\text{obs}})/u_{\text{err}})^2 \, \mathrm{d}\mathcal{A} + \frac{1}{2\mathcal{A}} \int ((v - v_{\text{obs}})/v_{\text{err}})^2 \, \mathrm{d}\mathcal{A}, \tag{6}$$

where $\mathcal{A} = \int \mathrm{d}\mathcal{A}$ is the total area, and $u$ and $v$ are the modelled horizontal $x$ and $y$ velocity components, respectively.

Úa employs Tikhonov regularisation, for which the regularisation term has the form

$$R_{\text{Úa}} = \sum_{k=1,2} \frac{1}{2\mathcal{A}} \int \left( \gamma_s^2 (\nabla(p_k - p_{k,\text{prior}}))^2 + \gamma_a^2 (p_k - p_{k,\text{prior}})^2 \right) \mathrm{d}\mathcal{A}, \tag{7}$$

where $\gamma_s$ and $\gamma_a$ are the slope and amplitude regularisation parameters, $p_1 = \log_{10} A$, $p_2 = \log_{10} C$ and $p_{k,\text{prior}}$ are prior values, or initial estimates, for the parameters $p_k$. For the inversions in this study, $\gamma_s = 1 \times 10^4$ m and $\gamma_a = 1$.

### 2.2.2 ISSM

In ISSM, the cost function is written as $\mathcal{J}_{\text{ISSM}} = I_{\text{ISSM}} + \alpha R_{\text{ISSM}}$, where $\alpha$ is the regularisation parameter. The misfit function is written as

$$I_{\text{ISSM}} = a I_{\text{abs}} + I_{\text{log}}, \tag{8}$$

where $a$ is a weighting parameter, adjusted such that the two components are equal in weight (within a given tolerance). $I_{\text{abs}}$ and $I_{\text{log}}$ are the absolute and logarithmic misfits given by

$$I_{\text{abs}} = \frac{1}{2} \int_s \left( (u - u_{\text{obs}})^2 + (v - v_{\text{obs}})^2 \right) \mathrm{d}s \tag{9}$$

$$I_{\text{log}} = \int_s \left( \log \left( \frac{\sqrt{u^2 + v^2} + \epsilon}{\sqrt{u_{\text{obs}}^2 + v_{\text{obs}}^2} + \epsilon} \right) \right)^2 \mathrm{d}s, \tag{10}$$

where $\epsilon$ is a minimum velocity applied to avoid numerical issues and $s$ is the ice surface.

The regularisation term is defined as

$$R_{\text{ISSM}} = \frac{1}{2} \int_{\Omega_p} \|\nabla p_k\|^2 \, \mathrm{d}\Omega_p, \tag{11}$$

where $\Omega_p$ refers to the ice volume with $p_1 = B$, or to the ice base with $p_2 = \beta^2$.

In ISSM, the inversions for each parameter are carried out independently of each other. First, $B$ is inverted for over a subdomain containing only the ice shelves, and then $\beta^2$ is inverted for on the grounded ice considering the whole domain, including the inverted value for $B$ on the ice shelves. The regularisation term takes different values in each step. For the $B$ inversion $\alpha = 1 \times 10^{-18}$, and for the $\beta^2$ inversion $\alpha = 1 \times 10^{-8}$.

### 2.2.3 STREAMICE

In STREAMICE, the parameters inverted for are $p_1 = \sqrt{B}$ and $p_2 = \beta$. The cost function is $\mathcal{J}_{SI} = I_{SI} + R_{SI}$. Since STREAM-ICE is not a purely finite element model, the functions are written discretely, taking the form

$$I_{SI} = \sum_{i,j \in D} \frac{1}{2N} \left( \frac{(u(i,j) - u(i,j)_{\mathrm{obs}})^2 + (v(i,j) - v(i,j)_{\mathrm{obs}})^2}{(1 + (u^2_{\mathrm{err}_{i,j}} + v^2_{\mathrm{err}_{i,j}})^{\frac{1}{2}})^2} \right) \tag{12}$$

$$R_{SI} = \sum_{k=1,2} \sum_{i,j \in D} \frac{1}{N} \gamma_k \left( \left( \frac{p_k(i+1,j) - p_k(i,j)}{\Delta x(i,j)} \right)^2 + \left( \frac{p_k(i,j+1) - p_k(i,j)}{\Delta y(i,j)} \right)^2 \right)$$

$$+ \sum_{i,j \in D_G} \frac{1}{N} \gamma_G (p_1 - B_0)^2, \tag{13}$$

where $i$ and $j$ are grid cell indices, $N$ is the total number of cells, $\Delta x$ and $\Delta y$ are the distances between grid cells in the $x$- and $y$-directions, $\gamma_1$, $\gamma_2$ and $\gamma_G$ are regularisation parameters, $B_0$ is an inital estimate for $\sqrt{B}$, $D$ is the full computational domain, and $D_G$ consists of the grounded cells only. Note that the summations in $I_{SI}$ and $R_{SI}$ are not weighted by cell area, as they would be for a discretely calculated domain integral (cf. $I_{\mathrm{ISSM}}$ and $I_{\mathrm{Úa}}$). This is in order to prevent the inversion from weighting larger cells too strongly. For the inversions in this work, $\gamma_1 = \gamma_2 = 2 \times 10^4$ and $\gamma_G = 1 \times 10^2$.

## 3 Experiment design and setup

### 3.1 Model domains and data

All three model domains used for our inversions, displayed in Fig. 2, cover both Thwaites and Pine Island glaciers, and extend west to include the Dotson and Crosson ice shelves. They are set up using bedrock and ice surface fields linearly interpolated from BedMachine Antarctica (Morlighem et al., 2019). STREAMICE includes a preprocessing step which applies a 5-pixel Gaussian smoothing filter to the surface and hydrostatically inverts for the regions of flotation and for the thickness of the ice shelf using the smoothed surface. It was found that without this smoothing, the inversion optimisation would stagnate in some cases (in particular, the experiment discussed in Appendix B4 using the initial guess referred to as 'Priors1'). The inversions use the surface velocities and measurement errors from the 2014-15 year of the updated dataset originally described in Mouginot et al. (2014). The same velocities and geometries are used throughout this study in all models. The densities are set to be constant and uniform, with values of $917 \, \mathrm{kg \, m^{-3}}$ for ice density and $1027 \, \mathrm{kg \, m^{-3}}$ for ocean water density, which are the values used in the hydrostatic equilibrium calculation for BedMachine.

The STREAMICE domain is a 528×720 cell rectangular grid, with a minimum resolution of 1 km at the centre of the domain, and maximum resolution at the edges of 5.4 km in the $x$-direction and 5.96 km in the $y$-direction. The other two models use triangular meshes with spatially varying resolution. Both have a finer resolution closer to the grounding line. The ISSM mesh contains 261,375 elements with edge lengths between 725 m on the ice shelf and 16 km in the coarsest areas, with a resolution of about 1 km close to the grounding line. The mesh was refined based on the distance from the grounding line and interpolation error of the observed ice velocity. The Úa mesh contains 213,828 elements with edge lengths varying linearly with the distance from the grounding line and additional refinement of areas with high velocity or strain rates, with resolution varying from 500 m to 15 km. The Úa mesh boundary was chosen based on the drainage basins of the glaciers and the location of the ice front, while the other two meshes cover larger areas which contain this region of interest. The triangular meshes used were created using BAMG (Hecht, 2006) in ISSM and Mesh2D (Engwirda, 2014) in Úa.

A Dirichlet boundary condition is used to set all velocities along the grounded parts of Úa's boundary to zero, since the boundary generally follows the edges of drainage basins. ISSM also uses Dirichlet boundary conditions, setting the velocities along the grounded parts of the boundary according to the velocity measurements. STREAMICE applies a no-flow boundary condition, as its boundaries are sufficiently far from the area of interest for this not to affect the outcome. All models apply the ice-front stress boundary condition along the seaward boundaries. In Úa, this is at the edge of the computational domain, and in the other two models the ice/ocean boundary is set using a mask derived from the BedMachine geometry data.

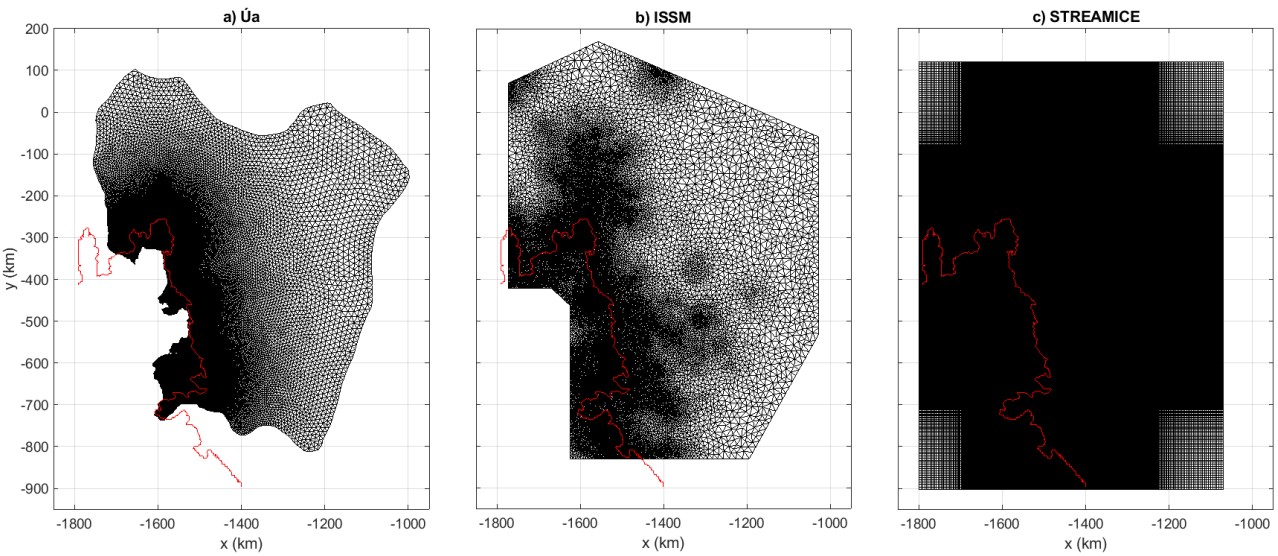

**Figure 2.** The meshes used by each model for the inversions. All domains cover our area of interest including Thwaites and Pine Island glaciers, and the Dotson and Crosson ice shelves. The main grounding line is shown in red.

In the time dependent simulations, surface mass balance is from a climatological record of RACMO2.1 (Lenaerts et al., 2012). For basal melting of the ice shelf we use the simple depth-based parameterisation

$$m_b = \begin{cases} 0 & \text{if } z \geq 0 \\ -\frac{75}{500}z & \text{if } 0 > z > -500 \\ 75 & \text{if } z \leq -500, \end{cases} \tag{14}$$

where $m_b$ is the basal melt rate in $\mathrm{m\,a^{-1}}$ and $z$ is the vertical coordinate in metres, positive upwards with zero at sea level.

## 3.2 Description of experiments

The first experiments involve a single inversion from each model. For the initial comparison, each model performs an inversion using the same geometry (bedrock and surface elevation) and velocity measurements, detailed in Sect. 3.1. In these experiments, the models are free to choose their own priors. All models calculate priors for $B$ using a relationship with temperature (e.g., Cuffey and Paterson, 2010), with Úa assuming a constant temperature of -15 °C, ISSM using the initial ISMIP6 temperatures (Seroussi et al., 2019) and STREAMICE using temperatures from Van Liefferinge and Pattyn (2013). The prior for the sliding coefficient in Úa is calculated from the Weertman sliding law using uniform values $v_b = 700\,\mathrm{m\,a^{-1}}$, $\tau_b = 80\,\mathrm{kPa}$ and $m = 3$. ISSM also computes a prior using the sliding law, but assumes $\boldsymbol{v_b}$ is equal to the observed velocities and the basal drag is equal to the driving stress. STREAMICE uses a uniform value of $150\,\mathrm{Pa^{1/2}\,m\,a^{-1/6}}$ as its initial guess for $\beta$.

The resulting fields of rate factor and basal sliding coefficients are compared directly in order to see whether the models produce similar results. The velocity misfits, defined as the difference between the modelled and observed values, are also compared as an indicator of how well the inversion processes have performed. The results of this comparison are found in Sect. 4.

Following this, further experiments seek to test the sensitivity of inversion outputs to particular details of the inversion procedure, such as the choices of optimisation scheme, algorithm sequence, mesh resolution and priors. An overview of the results of these experiments is found in Sect. 4.4, and more detail is available in Appendix B.

The final stage (Sect. 5) involves comparing the effects on the ice flow of using inversion outputs from each of the three models within transient runs. The $B$ and $\beta^2$ fields calculated by inversion are transferred between models and used as inputs. The models are run forward in time to investigate the effects of using outputs from different models' inversions on the evolution of the ice flow, with all else being equal.

## 4 Results of inversions

We first look at the outputs from inversions in the three ice flow models following the procedures previously described. The fields we compare are the speed misfit, and the values of $B$ and $\beta^2$. The outputs from all three models were converted to common units of $\mathrm{Pa\,a}^{\frac{1}{3}}$ for $B$ and $\mathrm{Pa\,m}^{-\frac{1}{3}}\,\mathrm{a}^{\frac{1}{3}}$ for $\beta^2$. These are the units used for all comparisons in this work.

For the purpose of the comparisons in this section, outputs from Úa and ISSM were interpolated linearly onto the rectangular grid of the STREAMICE domain. As can be seen from the shapes of the domains in Fig. 2, this results in some areas containing extrapolated values. In figures, these areas have been masked out to exclude extrapolated values, and all pairwise comparisons display only the region for which differences between directly calculated values are available. The ice mask for the STREAMICE domain has also been applied.

### 4.1 Speed misfit

Speeds $V = \sqrt{u^2 + v^2}$ and $V_{\mathrm{obs}} = \sqrt{u_{\mathrm{obs}}^2 + v_{\mathrm{obs}}^2}$ were calculated. The difference between modelled and observed speed, which we refer to as the misfit, is $V_{\mathrm{diff}} = V - V_{\mathrm{obs}}$.

The speed misfits for each model are displayed in Fig. 3. The speed misfit is a useful quantity to inspect in order to ensure that the inverted values of $B$ and $\beta^2$ produce reasonable velocities, but the exact magnitudes are not necessarily indicative of the quality of the inversions themselves. As shown in Sect. 2.2, the cost functions being minimised balance misfit and regularisation, thus different choices of regularisation in each inversion affect the misfit produced. The most important thing to note

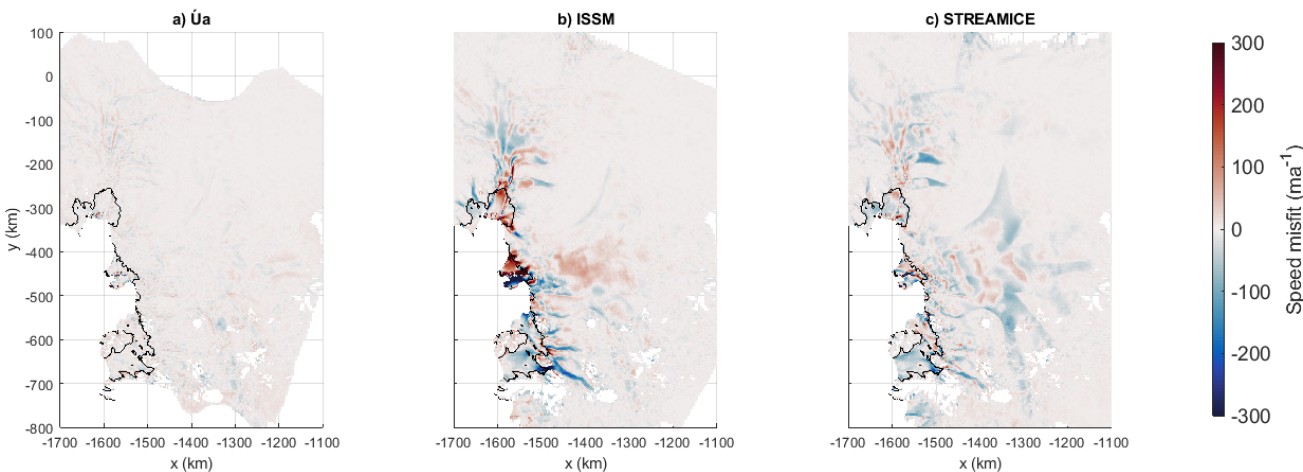

**Figure 3.** Difference in the calculated speeds after inversion, compared to the measurements, for each model. The grounding line is indicated in black.

| | Whole domain | Speed > 50 m a$^{-1}$ | Speed > 100 m a$^{-1}$ | Speed > 500 m a$^{-1}$ |
|---|---|---|---|---|
| Úa | 7.10 m a$^{-1}$ | 9.09 m a$^{-1}$ | 10.63 m a$^{-1}$ | 17.18 m a$^{-1}$ |
| ISSM | 19.43 m a$^{-1}$ | 35.39 m a$^{-1}$ | 49.77 m a$^{-1}$ | 104.03 m a$^{-1}$ |
| STREAMICE | 15.61 m a$^{-1}$ | 27.43 m a$^{-1}$ | 34.39 m a$^{-1}$ | 50.09 m a$^{-1}$ |

**Table 1.** Mean values for the magnitude of misfit in inversions from the three models, on regions with ice over chosen measured speed thresholds.

is that, following the inversions, calculated and measured velocities are similar for all three models.

A visual comparison reveals that Úa has minimised the difference furthest, with misfit under 50 m a$^{-1}$, except in localised spots such as the edge of the Thwaites Ice Tongue. The lower misfit compared to the other inversions can be attributed to the fact that Úa inverts for both $B$ and $\beta^2$ over the entire domain, and an experiment discussed in Appendix B2 shows higher misfit when this is not the case. The misfit of STREAMICE does not exceed 200 m a$^{-1}$ in general, again with a few small exceptions. ISSM displays higher misfits of hundreds of metres per year in certain locations, particularly on the Thwaites Ice Tongue. The higher misfit in ISSM could be due to the weighting of the absolute and logarithmic misfits in Eq.(8).

In general across all three models, the greatest differences are seen on the floating ice downstream of the grounding line, and on the fastest flowing grounded ice. For a clearer picture of the misfit on faster flowing ice, we can take the mean misfits on regions above a certain measured velocity threshold. The values for a few chosen thresholds are displayed in Table 1.

While we can see similarities in the locations of high misfit regions, the overall correlation between the distributions of misfit is not high. We calculated the Pearson correlation coefficient (Pearson, 1895) between each pair of misfit fields, and found the most similar to be Úa and STREAMICE, with a coefficient of 0.474. ISSM has a lower positive correlation with each of the other models, with coefficients of 0.276 and 0.270 for STREAMICE and Úa respectively. We note that these correlation coefficients serve only as rough quantitative estimates of the correlations between different inversion products. In general, we expect the correlation to depend on the spatial scales considered. For example, and as indicated by our inversion results, we generally observe better agreement over large spatial scales ($\geq 50$ km) than over smaller spatial scales.

## 4.2 Associated rate factor, $B$

The results from the rate factor inversion (Fig. 4) show the most widespread differences between the models. All the models produce values of similar magnitude, but the values in Úa are spread over a larger range and the field is less smooth. The smoother fields produced by ISSM and STREAMICE can be explained by the fact that these models do not generally invert for $B$ over grounded ice, as described in Sect. 2. Instead, both models calculate their priors from temperature, with STREAMICE using temperatures from Van Liefferinge and Pattyn (2013) and ISSM following the process described in Seroussi et al. (2019).

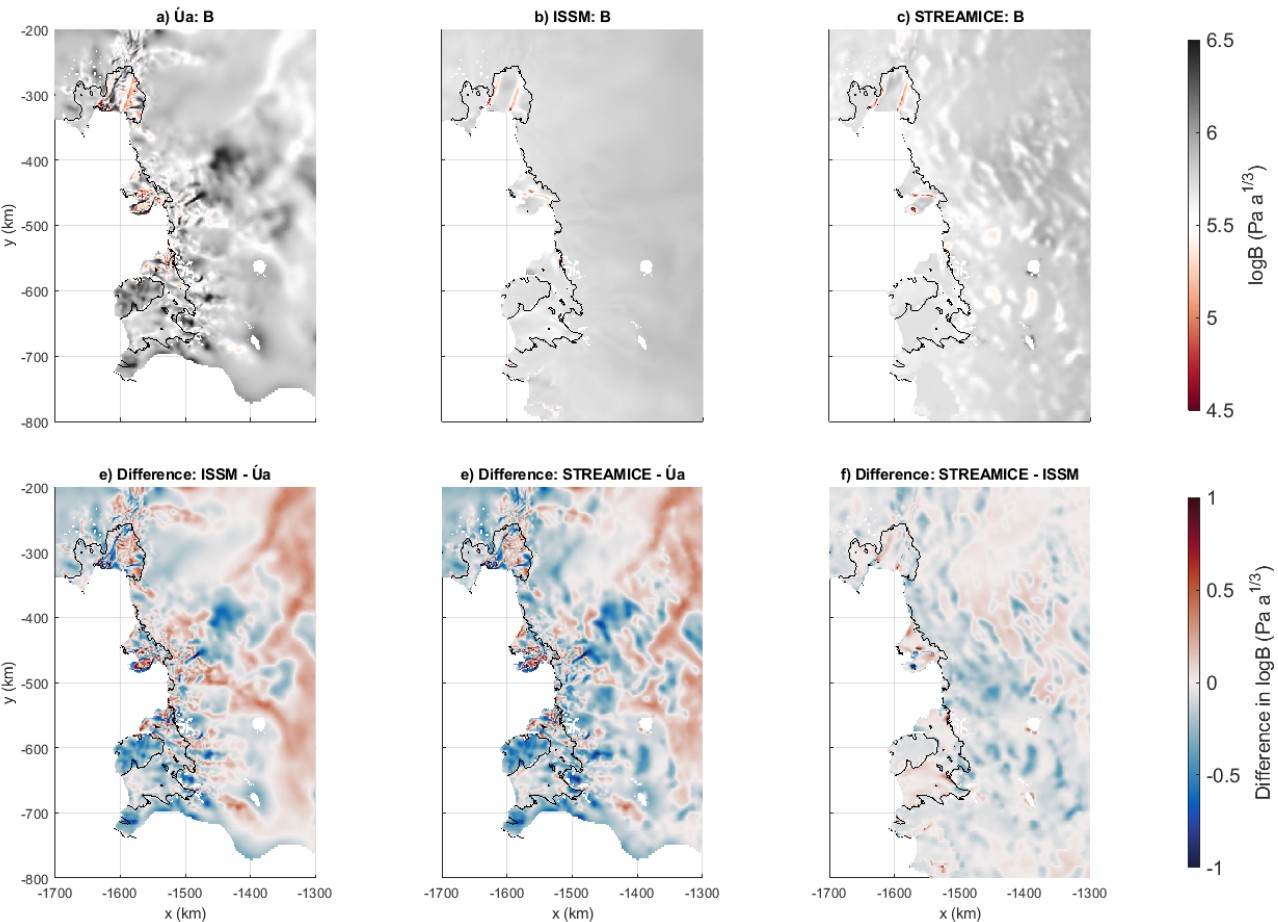

**Figure 4.** The $B$ fields calculated by inversion and differences between them, displayed on a logarithmic colour scale in units of $\mathrm{Pa}\,\mathrm{a}^{\frac{1}{3}}$. The grounding line is indicated in black.

STREAMICE does allow for some perturbation from the initial values on grounded ice if significant changes are needed to minimise the velocity misfit, but this is heavily restricted by the last term in Eq. (13). Meanwhile Úa, which allows optimisation of the rate factor over the entire domain, produces a much more spatially variable field over the grounded ice. This is likely

due in part to differences in regularisation applied in this particular example rather than a general feature. Locally, values of $B$ from Úa's inversion are up to an order of magnitude different from the prescribed temperature-based estimates used by the other two models.

On floating ice, ISSM and STREAMICE produce similar results, with differences between their outputs (Fig. 4(d-f)) gen-

erally being small. The Úa output differs from the other two, producing a more variable distribution over the ice shelves, as it does over the grounded sections. Úa's inversion produces softer ice on the Western ice shelf of Thwaites Glacier and close

to the calving front of Crosson Ice Shelf. However, it does also produce some similar features, with bands of softer ice being visible at the edges of the high-velocity ice streams which flow out onto Pine Island and Thwaites ice shelves. In general, the bigger differences are seen in faster-flowing areas, with the values for $B$ being most similar over Dotson Ice Shelf and the northern section of the Pine Island Ice Shelf, both of which have low measured surface velocities.

To provide some quantification of the differences between the rate factor fields calculated by the models, we use Pearson correlation coefficients as before. The coefficient values can be found in Table 2. Over the entire domain, the distribution produced by Úa is almost entirely uncorrelated with the output from the other two models. ISSM and STREAMICE, by contrast, are fairly well correlated, despite using different temperature fields to calculate the value on the grounded ice. When looking only at the floating ice, Úa shows a moderate positive correlation with the other models. This demonstrates a fairly significant effect of Úa performing the inversion for rate factor over the entire domain compared to the approaches of the other models.

## 4.3 Basal friction coefficient, $\beta^2$

Inverted $\beta^2$ fields (Fig. 5) show a greater agreement between models than the $B$ inversion products. This is likely because all three models are inverting for the parameter over the entire domain. An implication of this is that the inverted $\beta^2$ values appear not to be significantly dependent on the values of $B$, or on whether or not the two inversions are performed simultaneously.

However, there are still some notable differences between the $\beta^2$ fields. When compared to Úa, ISSM and STREAMICE have patches of lower $\beta^2$ values over the trunk of Pine Island Glacier which are less than $1 \, \mathrm{m}^{-\frac{1}{3}} \, \mathrm{a}^{\frac{1}{3}}$. Also, in all three difference plots (Fig. 5(d-f)), there are localised larger differences in the immediate vicinity of the grounding line. These are likely due to differences in how the grounding line is treated in the models, which are discussed in Appendix C.

The STREAMICE output occasionally contains "loops" of lower values. These can appear due to the model inverting for $\beta$ rather than $\beta^2$, and in some locations producing values of $\beta$ below zero, which is not physically viable. When $\beta^2$ is calculated from the final inversion output (a field of values for $\beta$), the shape of the function will be changed in any area where $\beta$ is negative to include peaks inside the rings of low values, rather than local minima.

Once again, calculating Pearson correlation coefficients between the outputs gives us a quantitative idea of how alike the distributions are. We find strong positive correlation coefficients in the region of 0.8 for each comparison pair (see Table 2 for exact values). This shows a high level of agreement between the $\beta^2$ outputs of different inversion processes, suggesting that the underlying model equations are well represented in these results, as opposed to the results being influenced by model-specific aspects of inversion processes.

|                                      | Úa & ISSM | Úa & STREAMICE | ISSM & STREAMICE |
|--------------------------------------|-----------|----------------|------------------|
| Speed misfit correlation             | 0.270     | 0.474          | 0.276            |
| $B$ correlation (whole domain)       | 0.077     | 0.058          | 0.666            |
| $B$ correlation (floating ice only)  | 0.368     | 0.340          | 0.511            |
| $\beta^2$                            | 0.843     | 0.871          | 0.798            |

**Table 2.** Pearson correlation coefficients calculated for different inversion outputs between all model pairs.

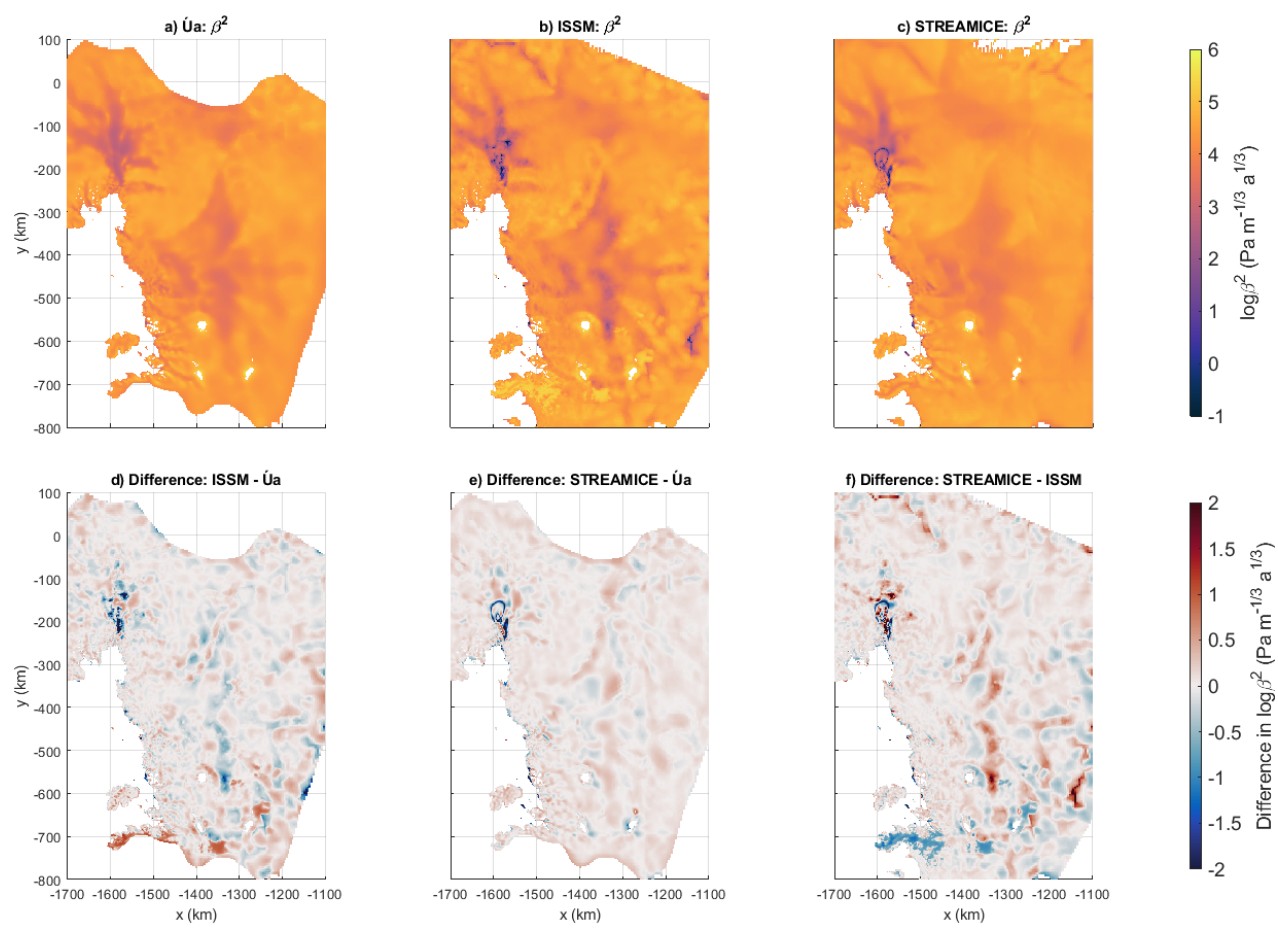

**Figure 5.** The $\beta^2$ fields calculated by inversion and differences between them, displayed on a logarithmic colour scale in units of $\mathrm{Pa\,m}^{-\frac{1}{3}}\,\mathrm{a}^{\frac{1}{3}}$. The ungrounded area of the domain has been masked out.

## 4.4 Discussion of inversion outputs

There are many factors which could cause differences in inversions. We have investigated several of these within Úa in an attempt to identify in particular why the difference in the misfit produced by Úa's inversions is lower than the other two models, and what causes patches of lower $\beta^2$ values to be produced over Pine Island Glacier. We summarise the findings here, but further details of this investigation can be found in Appendix B.

The difference in misfit appears to be due to a combination of factors. As noted in Sect. 4.1, it may just be a symptom of different regularisation choices between the models. However, we have identified other factors which may contribute to the difference. ISSM and STREAMICE use a different optimisation scheme, which results in a higher misfit when tested in Úa. However, the $B$ and $\beta^2$ fields from these experiments correlate well with Úa's original result, so the optimisation scheme does not account for the differences found in these outputs. Differences in meshes and the choice of priors were also tested and not found to cause significant changes to the inversion results, except in cases where the parameters were beyond the range of variation in our original inversions.

Major factors affecting the inversion results appear to be the section of the domain over which $B$ is inverted for, and the sequential nature of ISSM's inversion. Úa produces higher misfit when inverting only for $\beta^2$ with a predetermined $B$ field. In the $\beta^2$ field, patches of lower values are produced in a similar location over Pine Island Glacier to those noted in Sect. 4.3. The outputs from the experiment in which this was tested had lower correlations with the original inversion than those of any other experiments.

In general, the inversions were found to agree on large-scale distributions of $B$ on the ice shelves, and of $\beta^2$ everywhere. Due to our careful control of input datasets, we have removed much of the variability which can be introduced between models in general usage, and have shown the outputs to be robust with respect to technical aspects of the inversion process. An implication we take from this is that our inversion outputs are representative of the physics of the underlying equations rather than individual numerical details in the model code. Given the similarities in the inversion outputs, it may be possible to transfer them between models and recover broadly similar results in forward simulation. This is what we attempt next.

## 5 Transferring inversion outputs between models

We performed three time-dependent simulations in each of the three models, using a pair of inversion products from each model as inputs for the rate factor and basal sliding coefficient. The inversion outputs from Úa and ISSM are those used in the comparisons of Sect. 4, while the STREAMICE inversion outputs are from an inversion using slightly different priors. This is because, although the use of different priors does not greatly affect the spatial distribution of inversion outputs (as discussed in Appendix B4), it was found that forward runs in STREAMICE using its original inversion outputs encountered some conver-

gence issues.

The models were allowed to evolve for 40 years from the initial state described by our geometry datasets. The only differences between these simulations are the $B$ and $\beta^2$ fields. Additionally, in Úa, the value of $\beta^2$ was forced to be very low under ice which is initially floating, by setting $C = 1 \times 10^{10}$ (see Eq.(2)). This was done to discourage regrounding of ice, since there is a tendency in Úa for an initial grounding line advance to occur in the first few years of simulation.

Before running the full time-dependent simulations, we also looked at diagnostic simulations. However, we found that these were not a good indicator of the quality of inversions or forward model performance due to specific differences in the methods employed by our models at the grounding line. Details of this are given in Appendix C.

## 5.1 Results of time-dependent simulations

The changes in volume above flotation, ice mass and grounded area for the domain over 40 years are displayed in Fig. 6. The evolution of the ice sheet follows a similar trajectory in each model and with each pair of fields for rate factor and basal sliding coefficients, showing that inversion outputs can be transferred between models and produce reasonable results. However, the changes to ice volume and thickness over time happen at different rates depending on which model was used for the inversion, and in which model the simulation is run. Looking at the variance in volume above flotation across all nine experiments, the highest sea level contribution over the 40 years (the simulation run in Úa using ISSM's inversion outputs) is 42% greater than that of the lowest contributor (ISSM using Úa's inversion outputs), a difference of 4.8 mm over 40 years. The changes in total ice mass are closer, with a difference of 2434 Gt between the highest and lowest values, the largest change being 21% greater than the lowest.

Figure 7 displays the thickness changes over the 40 years of simulation, and the positions of the grounding line in each case. For the most part, similar patterns emerge in all the experiments, although there are some notable differences. While the distribution of thickness changes is similar, with the largest changes located at Thwaites Glacier and the Crosson Ice Shelf, the inversion outputs of STREAMICE generally cause faster thinning and grounding line retreat. The grounding line of Pine Island Glacier advances in the cases where ISSM and STREAMICE use the inversion outputs of Úa. This points to a discrepancy which could be rectified in future work to improve the consistency of results. The issue in this case is that the value of $\beta^2$ under the floating ice differs. Úa's inversion outputs contain higher $\beta^2$ values under ice which was originally floating, which causes regrounding to persist after any initial advance of the grounding line. If an alteration to the $\beta^2$ field were applied in ISSM and STREAMICE, as in Úa, it is likely that the agreement between them would be closer.

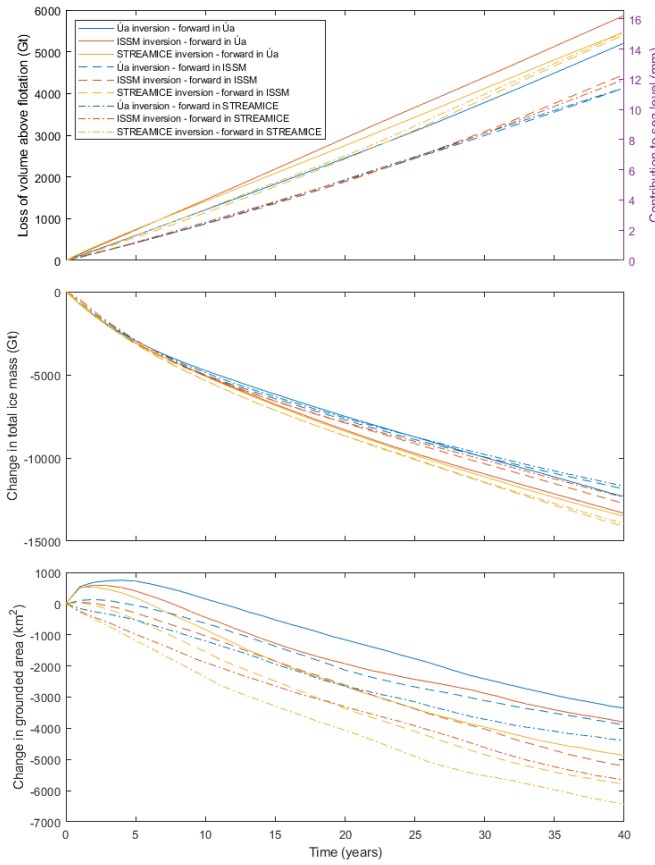

**Figure 6.** Changes in ice mass and grounded area over 40 years of simulation in Úa using the rate factor and basal sliding coefficient fields resulting from each of the three model inversions.

## 5.2 Discussion

Some differences between forward runs in different models, even when using the same inputs, are to be expected, as model intercomparisons demonstrate (e.g., Bindschadler et al., 2013; Asay-Davis et al., 2016; Cornford et al., 2020). This could be due to many factors. For example, our models use different meshes, and are solving different stress balances, with STREAM-ICE using an L1L2 approximation while ISSM and Úa use SSA. In our experiments, we find differences in behaviour at the grounding line (as detailed in Appendix C) to be one particular cause of differing behaviour between the models. As an example of this, we can see that in Úa's simulations the grounded area (Fig. 6c) increases a little before following the same downward trend as the other two models. Additionally, in Fig. 7, we see that ISSM's forward runs consistently show the grounding line of Thwaites Glacier retreating more than in the other two models.

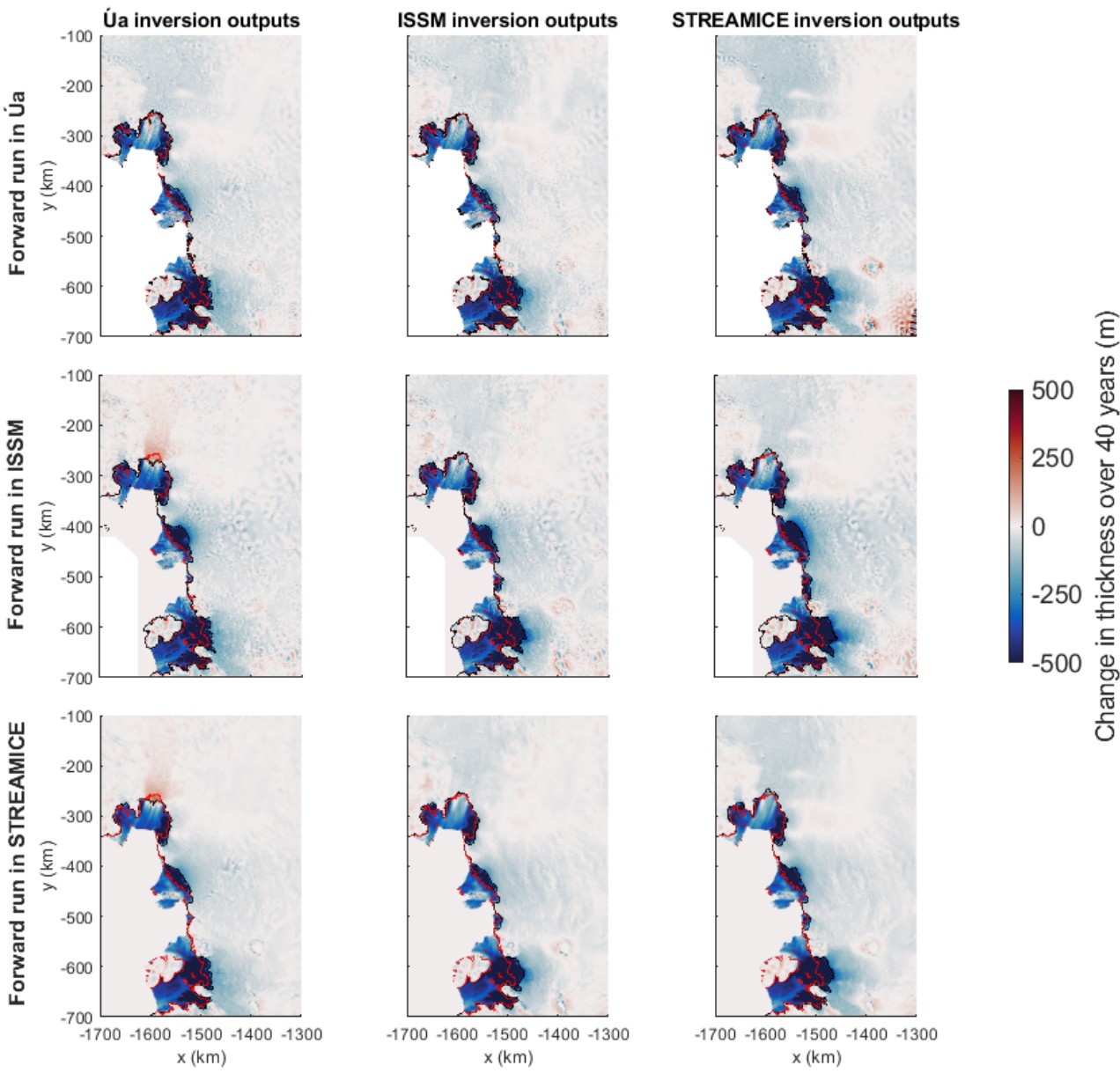

**Figure 7.** Thickness changes after 40 years of simulation using the rate factor and basal sliding coefficient fields resulting from each of the three model inversions. The initial grounding line position is indicated in red, and the final position in black.

Our aim was to investigate the transferability of inversions between models, and for this we examine the results in sets of three, comparing the outcomes produced in the same forward model using the three different inversion outputs. In Úa, the range of sea level contribution after 40 years resulting from the three sets of inversion outputs is 1.82 mm, with the highest contribution being less than 13% higher than the lowest. In ISSM, the forward simulations produced a 3.52 mm range in sea level contributions, representing a 31% difference. In STREAMICE, these values were 3.67 mm and 32%. The result from Úa particularly highlights the potential for successful transfer of inversion products. As noted above, it is likely that the variation in ISSM and STREAMICE would be lower if the $\beta^2$ values under initially floating ice were controlled, as they were for Úa's simulations.

It should be noted that, again using sea level contributions as the metric for comparison, the largest difference between the results from each model under normal usage (i.e. all models using their own inversion outputs) is 2.78 mm, which is within the range of variability we see when using different inversion outputs within each of the three models. This means that the observed differences between the model outputs appear to be similarly affected by the inversion products used as they are by the model used.

Another important note to emphasise is that the low variability in our time-dependent results demonstrates that the exact magnitude of misfit is not a direct reflection of the quality of an inversion. Higher misfit values were observed in the ISSM inversions (Sect. 4.1), but this did not result in any large differences in forward runs compared to inversions from the other two models.

## 5.3   Comparisons with previous studies

After 40 years, the contributions to sea level calculated in our three models using the three sets of model inversions differ by up to 4.8 mm, a factor of 1.42. The largest variation calculated within the same model is 3.67 mm, a factor of 1.32. Although seemingly a fairly large difference, this can be seen as a good result in the context of model initialisation processes. This range compares favourably to the control experiment of initMIP-Antarctica (Seroussi et al., 2019), where climate forcing remains constant, which found a range of sea level contributions between -243 mm and +167 mm for the whole of Antarctica over 100 years. This included models which use entirely different initialisation methods and datasets, but the two extreme values are both from models which utilise types of inversion. Compared to our experiments, there are many additional factors which could cause differences between the initMIP-Antarctica results. However, within this range are examples of simulations more comparable to our own, which differ greatly despite being run in the same model. Two simulations using ISSM differed by a factor of 3, with sea level contributions of -80.7 mm and -243.6 mm. Both ISSM variants inverted for basal sliding, but used different datasets in their inversions.

The variability in ice mass loss found in our experiments is less than that of the control experiment of Alevropoulos-Borrill et al. (2020), in which sensitivity of simulations to basal sliding, rate factor and basal melt were tested by using perturbed

parameter fields. The low- and high-end parameter sets produced sea level contributions of 0.03 mm and 28.85 mm for the Amundsen Sea Embayment. These parameter sets were based on the 5th and 95th percentiles of the probability density of an ensemble of experiments from Nias et al. (2016), in which they produced average sea level rise of $0.002\,\mathrm{mm\,a^{-1}}$ and $0.682\,\mathrm{mm\,a^{-1}}$ respectively over a 50 year simulation. The results of our experiments are comfortably in the middle of this range, showing that the uncertainty introduced by transferring our models' inversions is much smaller than this parameter uncertainty.

Our results display lower variability in terms of sea level contribution than that seen between models in the Antarctic intercomparison of the SeaRISE project (Bindschadler et al., 2013) or the results for the Amundsen Sea sector in LARMIP-2 (Levermann et al., 2020). The variability is also lower than the experiments of Favier et al. (2014), in which three models of varying complexity are compared. The differences between values for contribution to sea level also fall within the ranges reported in studies assessing other aspects of models. The comparisons here are less direct, since model process studies can entail changes to the physics of models whereas our forward experiments only deal with changes to inputs. However, we suggest that the uncertainties introduced by using inversion products outside of their native model are not more than those of, for example, the different climate forcings used in LARMIP-2 (Levermann et al., 2020) or the choice of sliding law in Yu et al. (2018) or Brondex et al. (2019).

These favourable comparisons demonstrate the value of the standardisation of input datasets in our inversions, which helps to minimise uncertainty when transferring them. As long as the same geometry and densities are used as in the inversion process, no more uncertainty is introduced into a forward problem by choosing to use an inversion product from another model than by other standard modelling choices such as which sliding law to use.

## 6 Conclusions

In this work, we have investigated the differences between inversions for flow rate factor and basal sliding coefficients calculated in three different ice flow models. They each use different inversion equations and techniques, but despite this they display a high degree of agreement in patterns of distribution, with strong positive correlations particularly evident between the fields of basal sliding coefficients. The implication of this is that outputs of inversions contain minimal representation of model-specific numerical behaviour, and strongly reflect the underlying equation system the models are designed to solve. The results of inversion processes used by our models are shown to be consistent with each other to a higher extent than may have been expected from the ill-posedness of the problem being solved. The minimal model-dependence demonstrates that ice flow models are as robust in their inversions as they are in their forward simulations.

Further to this, we have shown that the products of inversions performed in any one of these three models can be used in any of the two other models as an input for transient simulations, and that the results obtained this way are similar to those obtained

when each model uses its own inversion products. Hence, the inversion products can be described as transferable between models. In our 40 year transient simulations, the variation in sea level contributions produced by a single model did not exceed 32%, and further efforts to standardise modelling procedures would likely improve this figure. The smallest variation found between simulations using the three different sets of inversion products was 13%. We found that using inversion products from different models results in similar variability to that which already exists between each of the models operating normally with their own inversion outputs.

Due to our careful control of input datasets, the results of our time-dependent simulations show variability lower than those of other intercomparison experiments. When the process is managed well, the variability introduced by transferring inversion outputs from one model into another is not significantly high, and thus is not prohibitive to wider applications. With provision of sufficient details of the models involved, it would be possible to produce fields of basal sliding coefficients and rate factors which could be used by multiple models for the purpose of increasing uniformity in the boundary conditions and ice properties of intercomparison projects, or could be used as inputs for models which cannot perform their own inversion calculations.

*Code availability.* Source code for Úa can be downloaded at http://doi.org/10.5281/zenodo.3706623, and requires MATLAB to run. Source code for ISSM can be downloaded at https://issm.jpl.nasa.gov/download. STREAMICE is part of MITgcm, for which the source code is found at http://mitgcm.org/source-code. The OPTI Toolbox for MATLAB, containing M1QN3, can be downloaded at https://www.inverseproblem.co.nz/OPTI/.

**Appendix A: Regularisation of inversions**

The choice of regularisation parameters in our inversions is based on L-curve analysis. L-curves (e.g., Hansen, 1992) are created by plotting the regularisation against the misfit ($R$ against $I$, from the general form of inversions given in Eq.(5)). The misfit is generally reduced by increasing the regularisation, but the relationship is not linear. The plots follow an L-shape, and the aim is to pick parameters whose results lie near the corner of the L, where neither $I$ nor $R$ take high values. This ensures that the the inversions produce velocities close to the measurements without being over-regularised.

In Úa, there are technically four different regularisation parameters. In Eq.(7), we use the notation $\gamma_s$ and $\gamma_a$, but it is possible for these to be set differently for each $p_k$. In practice, we use the same values of these parameters for both $A$ and $C$, hence the notation in the equation. From experience, we know that values of $\gamma_s = 1 \times 10^4$ and $\gamma_a = 1$ are a good starting point. These values were confirmed as good choices by the L-curves produced from a series of tests, shown in Fig. A1.

In ISSM, since the inversions for $B$ and $\beta^2$ are carried out separately, the parameters are also chosen separately in two L-curves. The values for the $B$ inversion are chosen first, and once the optimal values are selected, the corresponding $B$ field is carried forward into the L-curve analysis for $\beta^2$. The L-curves used to choose the regularisation values in this project are

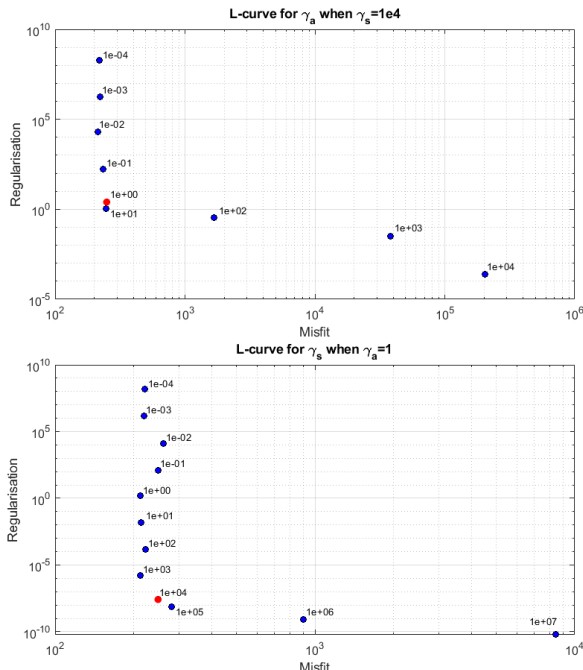

**Figure A1.** L-curves for Úa. Points are labelled with values of $\gamma_a$ and $\gamma_s$, and the chosen values are indicated in red.

displayed in Fig. A2.

The regularisation parameters for STREAMICE were initially chosen to be the values resulting from previous work (Goldberg et al., 2019). An L-surface produced for that work is displayed in Fig. A3. These values were then adapted slightly to improve optimisation performance for this project.

## Appendix B:  Investigating the effects of differing aspects of model inversion processes

After observing the differences between the inversion results of the three ice flow models, we investigated possible causes for them. Each of the models approaches the inversion process in a slightly different way, and further testing would reveal which factors are the most influential in affecting the outcome. We tested different factors by performing independent inversion calculations for each case in Úa, and in one case across all three models. We looked at the velocity misfits, rate factors and basal sliding coefficients produced as indicators of inversion performance compared to the original results. We attempted to determine from this how robust our inversion results are with respect to these procedural differences.

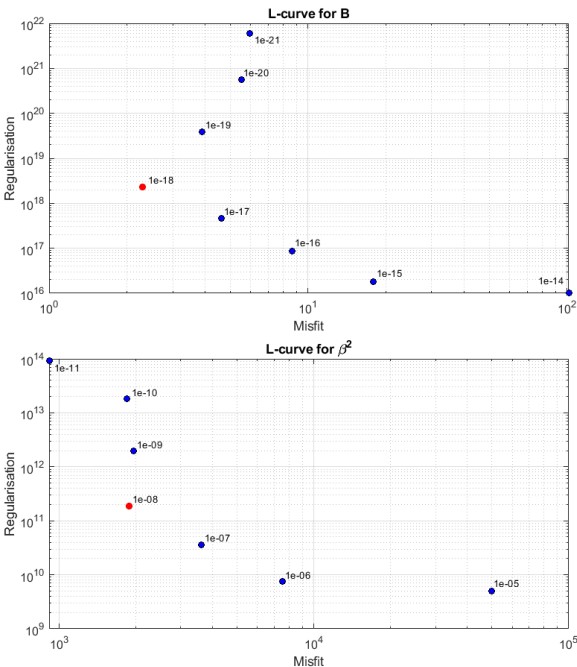

Figure A2. L-curves for ISSM. Points are labelled with values of $\alpha$, and the chosen values are indicated in red.

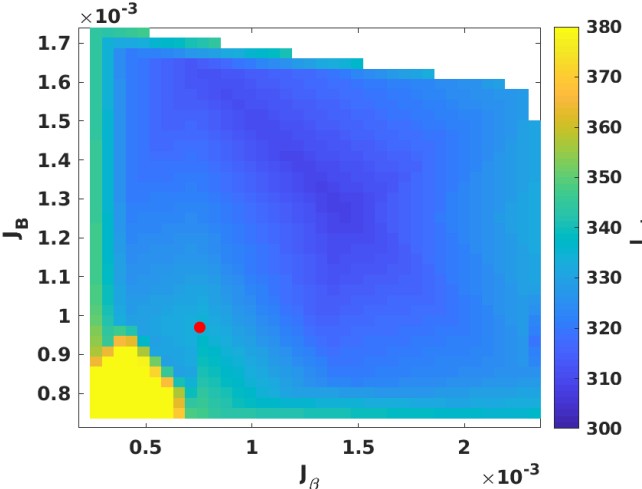

Figure A3. An L-surface for STREAMICE, produced for a previous project, which was used as the basis for regularisation choices in this work. The chosen values lie at the red point.

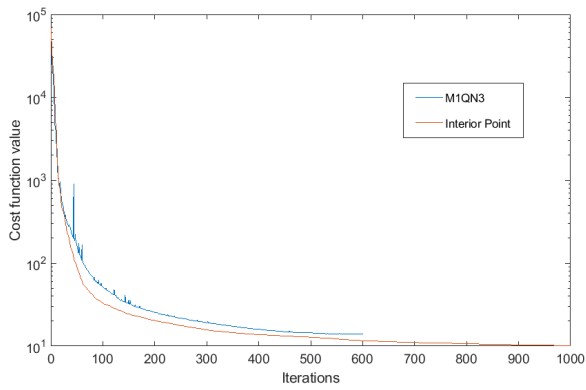

**Figure B1.** A comparison of the performance of the Interior Point algorithm in MATLAB used by default in Úa, and the M1QN3 optimisation scheme used by ISSM and STREAMICE, showing minimisation of the cost function during the inversion process.

## B1 Optimisation schemes

One possible source of inconsistency between the models is the optimisation scheme used during the inversion process. ISSM and STREAMICE both make use of a scheme called M1QN3 (Gilbert and Lemaréchal, 1989), while Úa uses the Interior
Point algorithm (Byrd et al., 1999) via MATLAB's inbuilt 'fmincon' function. Since Úa appeared to be performing better in minimising the differences between modelled and observed velocities, this was an important operational difference to check. We compared these algorithms in Úa by using a MATLAB implementation of M1QN3 from the OPTI Toolbox (Currie and Wilson, 2012). Used on the same inversion problem, M1QN3 under-performed compared to Interior Point algorithm. The inversion process aborted after 601 iterations, with the cost function having converged to a minimum value of 13.96, while
the Interior Point algorithm reduced the cost function to 11.64 in the same number of iterations. The Interior Point algorithm continued to further minimise the cost function until the process was stopped at the 1000 iteration limit set in Úa, at which point the cost function value was 10.12. The minimisation processes for both algorithms are shown in Fig. B1.

The misfit fields resulting from these inversions (Fig. B2(b)) do not show a great enough difference in magnitude to entirely
account for the discrepancies observed between the inversions from different models. While the M1QN3 inversion is visibly performing less well than MATLAB's Interior Point scheme, the misfit is smaller than those seen in results from the two models using M1QN3 by default. The mean magnitude of misfit using M1QN3 in Úa is $9.61 \, \mathrm{m \, a^{-1}}$, compared to $7.10 \, \mathrm{m \, a^{-1}}$ using the Interior Point scheme. By this measure it could account for roughly 30% of the difference observed between Úa and STREAMICE in Sect. 4.1, and 20% of the difference seen in ISSM. In this case, as opposed to in the original inversion
comparisons, the misfit is a more direct indicator of performance. Since the regularisation is exactly the same in both cases, the only differences in misfit are due to the choice of optimisation scheme.

It is interesting to note that use of the M1QN3 algorithm results in slightly lower values of $\beta^2$ on part of Pine Island Glacier, in a similar location to the low-value patches seen on the ISSM and STREAMICE inversions. The earlier termination of the minimisation process compared to Úa's Interior Point algorithm could be a cause of differences in that area. On the whole, however, there is a strong positive correlation in the spatial distribution of both the basal sliding coefficients and the speed misfit when comparing the two optimisation algorithms, with a weaker correlation in the rate factor. The correlation coefficients for several experiments described in this appendix can be found in Table B1.

### B2   Inverting for basal sliding alone

The way in which ISSM performs its $B$ and $\beta^2$ inversions sequentially rather than simultaneously and, in common with STREAMICE, does not generally invert for $B$ over the grounded ice, could impact the result. To test this, we took the result of the $B$ inversion from ISSM and used it as an input for an inversion in Úa. In this experiment, Úa was used to invert only for basal sliding coefficient, without changing the rate factor.

The results of this test (Fig. B2(c)) show that using ISSM's calculated $B$ field causes larger misfits in the velocity of the floating ice, especially on the Thwaites Ice Tongue. This is consistent with differences seen in the original misfit comparison (Sect. 4.1). Some of those differences also propagate upstream of the grounding line. Fixing $B$ to the values calculated from a temperature field causes a patch of low $\beta^2$ values to form over Pine Island Glacier, in a similar way to the original results from the ISSM and STREAMICE inversions. The values dip below $10^2 \, \mathrm{Pa} \, \mathrm{m}^{-\frac{1}{3}} \, \mathrm{a}^{\frac{1}{3}}$, but do not reach the lowest values found in the original results. It appears that Úa's usual method of inverting for both parameters across the entire domain causes information which would otherwise be interpreted as extreme lows in the $\beta^2$ field to be absorbed into the $B$ values instead. This can explain some of the differences seen in the original inversions.

The correlation of this $\beta^2$ distribution with the outputs of the original Úa inversion is very low, despite a visual inspection of the results showing similarities and familiar features. This is caused by localised spikes of extreme values affecting the calculation. Once the $\beta^2$ field is edited to remove these extreme values, by capping values at $1 \times 10^6 \, \mathrm{Pa} \, \mathrm{m}^{-\frac{1}{3}} \, \mathrm{a}^{\frac{1}{3}}$, the correlation coefficient is recalculated as 0.512. This is in the region which would be expected from visual inspection of Fig. B2, although still a weaker correlation than those found for other factors under investigation. The issue of localised extreme values affecting the correlation is one which is also encountered for the $B$ field, and in further results of this appendix. They are indicated in Table B1.

### B3   Mesh and resolution

The models are performing their calculations over different meshes, so experiments to test the mesh-dependence of inversions were performed. We first tested the mesh of ISSM within Úa, and found that the inversion outputs are not particularly sensitive to the location of mesh points if the resolutions are similar as is the case here. A comparison produced a strong positive

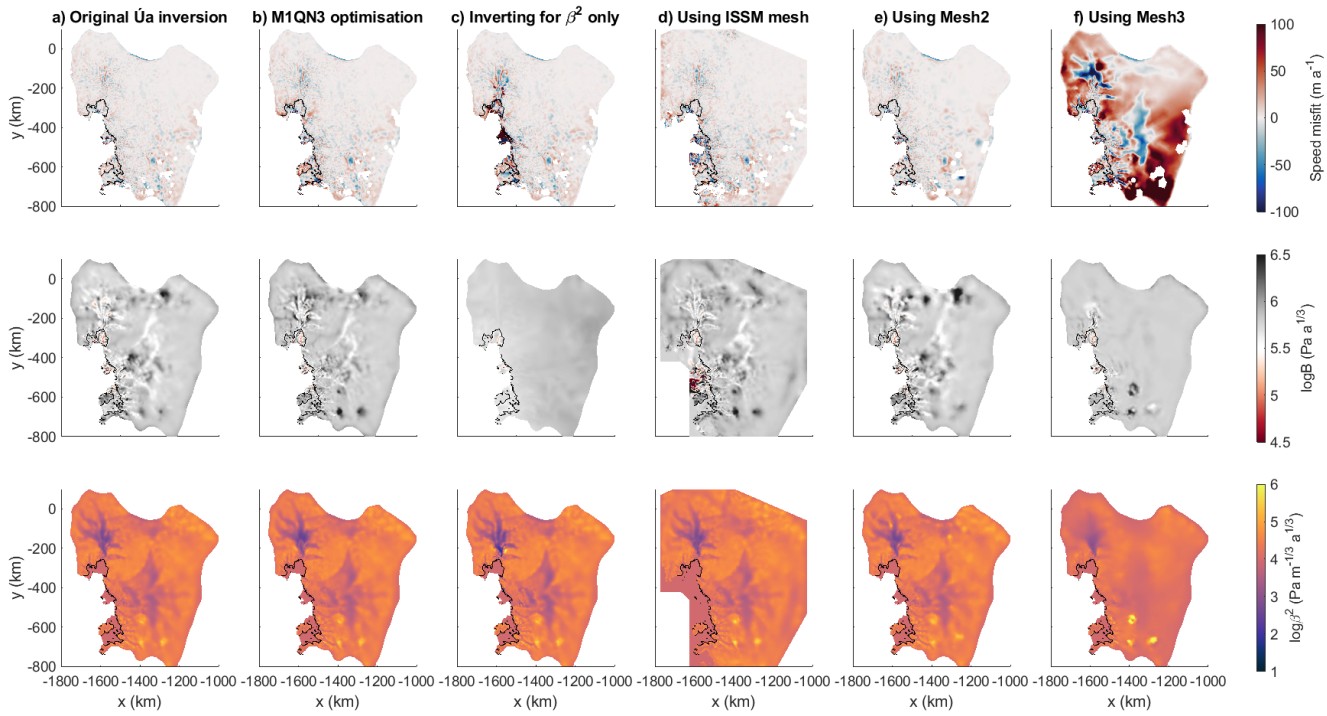

**Figure B2.** A comparison of the speed misfit, rate factor and basal sliding coefficients for several cases of inversions in Úa under different conditions. Column a) The original Úa inversion for $B$ and $\beta^2$ using an Interior Point optimisation scheme. Column b) The inversion using M1QN3 optimisation scheme. Column c) Inverting for $\beta^2$ only, using ISSM's $B$ field. Column d) Using the mesh and domain of the original ISSM inversion. Column e) Using the second (coarser) version of Úa's mesh. Column f) Using the third (even coarser) version of Úa's mesh.

|  | ISSM | STREAMICE | M1QN3 | $\beta^2$ only | ISSM mesh | Mesh2 | Mesh3 |
|---|---|---|---|---|---|---|---|
| Speed misfit correlation | 0.270 | 0.474 | 0.980 | 0.021 | 0.743 | 0.963 | 0.863 |
| $B$ correlation (whole domain) | 0.077 | 0.058 | 0.449 | 0.076* | 0.389* | 0.220* | 0.238* |
| $B$ correlation (floating ice only) | 0.368 | 0.340 | 0.455 | 0.116* | 0.350* | 0.206* | 0.236* |
| $\beta^2$ correlation | 0.843 | 0.871 | 0.914 | 0.512† | 0.929 | 0.945 | 0.552† |

**Table B1.** The Pearson correlation coefficients of various models and tests with Úa's original inversion. The first two columns show the correlation of the original ISSM and STREAMICE inversions, and the remaining columns show the correlation with the cases displayed in Fig. B2(b-f). *Values limited to $5 \times 10^7$ Pa a$^{\frac{1}{3}}$ before calculating correlation. †Values limited to $1 \times 10^6$ Pa m$^{-\frac{1}{3}}$ a$^{\frac{1}{3}}$ before calculating correlation.

correlation in the $\beta^2$ and misfit distributions, with Fig. B2(d) showing that there is not a large difference in the velocity misfit. The results for $B$ show greater variation.

The minimum length of the elements in Úa's original mesh is $500\,\mathrm{m}$, but STREAMICE uses a mesh with minimum resolution of $1\,\mathrm{km}$. STREAMICE uses rectangular elements in its mesh which Úa cannot replicate, so instead we looked at the effects of changing the mesh resolution. Performing the same inversion in Úa over different resolutions shows some interesting results. For these experiments, we used coarser versions of the Úa mesh, which we refer to as 'Mesh2' and 'Mesh3'. These have the same boundary, but are designed to have element edge lengths two and three times those of the original Úa mesh. Thus, Mesh2 contains 58,292 elements with edge lengths between $1\,\mathrm{km}$ and $30\,\mathrm{km}$, and Mesh3 contains 30.421 elements with edge lengths between $1.5\,\mathrm{km}$ and $45\,\mathrm{km}$. An inversion over Mesh2 produces a slightly larger misfit (Fig. B2(e)) than the original mesh, but the distribution of the misfit field correlates strongly with that calculated on the original mesh. There is also a high correlation between the $\beta^2$ fields produced in each case, with the differences between the two cases being primarily apparent in the $B$ field.

Using Mesh3 (Fig. B2(f)), we see a much greater difference. In this case the misfit is far higher, and there are noticeable differences in the fields of $B$ and $\beta^2$, which are lacking much of the detail present when using the other meshes. This shows that there is a limit to the mesh resolution from which useful inversion results can be obtained using the same parameters and model settings.

Within the range of resolutions of our models in their original states, the inversions for $\beta^2$ are robust and consistent. However, the results of the experiment using Mesh3 show that inversions performed on meshes with significantly lower resolutions do not retain this consistency. The inversions for $B$ appear to be more mesh-dependent, with far lower correlation.

## B4 Priors

In the original inversion comparison, each model was given the freedom to pick its own default priors for $B$ and $\beta^2$. This choice of a starting point for the inversion could have an effect on the outcome. To test this, two inversions were run in each model with identical priors. One set of priors (Priors1) consists of uniform values for $B$ and $\beta^2$, such that $\log(B) = 5.7$ and $\log(\beta^2) = 4.3$. The other (Priors2) consists of ISSM's original prior for $B$, and a $\beta^2$ field calculated using the Weertman sliding law with our velocity dataset and a constant value of $\tau_b = 80\,\mathrm{kPa}$.

The results in Fig. B3 show that all models have some dependency on the priors chosen. In general, the rate factor is most heavily influenced by the prior field used, due to two of the models not changing these initial values over much of the domain. Even in Úa, which does invert for $B$ everywhere, the influence of the prior values can clearly be seen on the slow-flowing ice inland. Pearson correlation coefficients for $B$ are found in Table B2. Looking at the floating ice only, using the same priors across all three models does not have a significant effect on the correlation of $B$ distribution between Úa and either of the other models. However, there is a noticeable strengthening of the correlation between the inverted values of ISSM and STREAMICE

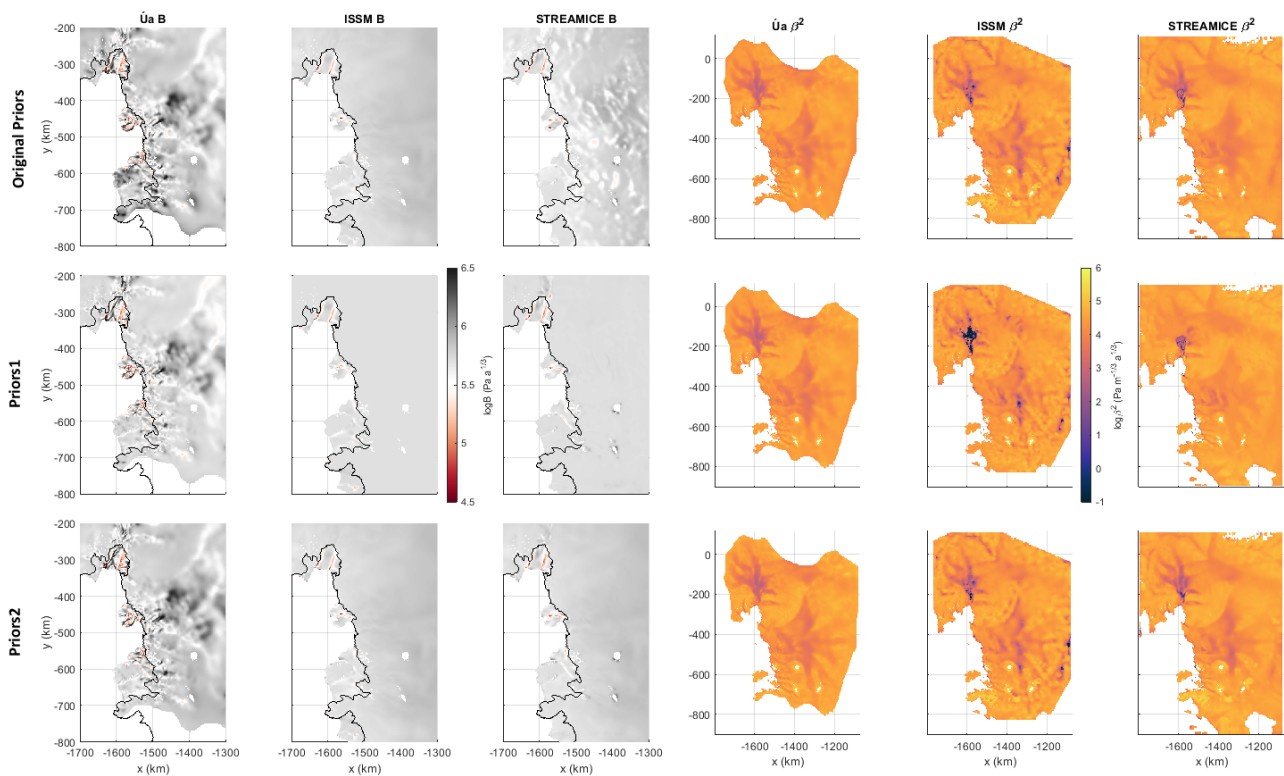

**Figure B3.** A comparison of the outputs of inversions using the original priors from each model in the first row, and the two specified sets, Priors1 and Priors2, in the second and third rows.

| | Úa | | | ISSM | | | STREAMICE | | |
|---|---|---|---|---|---|---|---|---|---|
| | Original | Priors1 | Priors2 | Original | Priors1 | Priors2 | Original | Priors1 | Priors2 |
| Úa Original | | 0.794 | 0.793 | 0.077 | 0.204 | 0.077 | 0.058 | 0.274 | 0.127 |
| Úa Priors1 | 0.817 | | 0.772 | 0.099 | 0.175 | 0.099 | 0.056 | 0.219 | 0.134 |
| Úa Priors2 | 0.794 | 0.864 | | 0.424 | 0.226 | 0.424 | 0.317 | 0.299 | 0.455 |
| ISSM Original | 0.368 | 0.365 | 0.353 | | 0.526 | 1.000 | 0.666 | 0.424 | 0.932 |
| ISSM Priors1 | 0.374 | 0.358 | 0.329 | 0.901 | | 0.526 | 0.238 | 0.626 | 0.434 |
| ISSM Priors2 | 0.368 | 0.365 | 0.353 | 1.000 | 0.901 | | 0.666 | 0.424 | 0.932 |
| STREAMICE Original | 0.340 | 0.334 | 0.313 | 0.511 | 0.550 | 0.511 | | 0.399 | 0.726 |
| STREAMICE Priors1 | 0.433 | 0.423 | 0.387 | 0.657 | 0.729 | 0.657 | 0.794 | | 0.583 |
| STREAMICE Priors2 | 0.455 | 0.353 | 0.423 | 0.712 | 0.697 | 0.712 | 0.744 | 0.941 | |

**Table B2.** The Pearson correlation coefficients for $B$ between pairs of tests using different sets of priors. Above the diagonal are coefficients calculated over the entire domain. Below the diagonal are the coefficients calculated over the floating ice only.

| | Úa | | | ISSM | | | STREAMICE | | |
|---|---|---|---|---|---|---|---|---|---|
| | Original | Priors1 | Priors2 | Original | Priors1 | Priors2 | Original | Priors1 | Priors2 |
| Úa Original | | 0.989 | 0.970 | 0.843 | 0.715 | 0.819 | 0.871 | 0.732 | 0.885 |
| Úa Priors1 | 0.953 | | 0.984 | 0.838 | 0.722 | 0.813 | 0.864 | 0.729 | 0.883 |
| Úa Priors2 | 0.952 | 0.956 | | 0.835 | 0.698 | 0.808 | 0.850 | 0.726 | 0.886 |
| ISSM Original | 0.270 | 0.264 | 0.265 | | 0.749 | 0.971 | 0.798 | 0.669 | 0.867 |
| ISSM Priors1 | 0.217 | 0.219 | 0.210 | 0.747 | | 0.748 | 0.681 | 0.520 | 0.688 |
| ISSM Priors2 | 0.280 | 0.272 | 0.273 | 0.958 | 0.755 | | 0.779 | 0.664 | 0.847 |
| STREAMICE Original | 0.474 | 0.463 | 0.463 | 0.276 | 0.220 | 0.274 | | 0.815 | 0.904 |
| STREAMICE Priors1 | 0.324 | 0.321 | 0.305 | 0.186 | 0.118 | 0.148 | 0.407 | | 0.781 |
| STREAMICE Priors2 | 0.633 | 0.630 | 0.639 | 0.329 | 0.249 | 0.324 | 0.687 | 0.428 | |

**Table B3.** The Pearson correlation coefficients for $\beta^2$ and speed misfit between pairs of tests using different sets of priors. Above the diagonal are coefficients calculated for $\beta^2$. Below the diagonal are the coefficients calculated for speed misfit.

over the ice shelves using both Priors1 and Priors2.

In the $\beta^2$ results, we find a slightly different outcome. Úa's results are affected very little by changing the priors, whereas the original outputs from ISSM and STREAMICE show a greater correlation with the outputs using Priors2 than for Priors1. The difference can be seen in Fig. B3 most prominently over Pine Island Glacier, where the extreme low values of $\beta^2$ in ISSM are exaggerated further using Priors1, while the distribution in STREAMICE does not capture the tributaries feeding into the main trunk of Pine Island. This is likely due to the values of $B$ under the ice. For Priors1, they take a uniform value not based on a temperature distribution, so greater changes to $beta^2$ are necessary to minimise the calculated velocity misfit. The results tell us that Úa is the least sensitive model to a change in priors, which we postulate to be due to the model inverting for both $B$ and $\beta^2$ over the whole domain.

It may well be the case that only the choice of prior for $B$ has a significant effect on the outputs in ISSM and STREAMICE, due to the fact that the chosen value will not be changed by the inversion processes. The choice of priors appears only to be a matter for concern in if a reasonable field cannot be calculated from existing velocity or temperature data.Even the lowest correlation for $\beta^2$, shown in Table B3, between ISSM and STREAMICE both using Priors1, is greater than 0.5.

The strong correlations between each model's original output and the Priors2 experiments leads us to conclude that the choice of priors is not a major factor in the differences between the original inversions, as neither STREAMICE nor ISSM was using uniform priors.

While the priors do not appear to affect the inversion outputs greatly, it was found that forward runs in STREAMICE using its original inversion outputs encountered some convergence issues, while this was not the case with the outputs from inversion using Priors2. For this reason, the Priors2 inversion from STREAMICE is used in the forward runs of Sect. 5.

### B5    Derivation of the adjoint

A technical difference between the inversion procedures in our models is the derivation of the adjoint. Úa and ISSM use an exact adjoint, following the terminology of Morlighem et al. (2013). STREAMICE uses the method described in Goldberg et al. (2016), but with a relatively weak tolerance which places it somewhere between the exact and incomplete adjoints. A straightforward comparison is difficult here as STREAMICE is using a different stress balance, since its equations use an L1L2 approximation (Goldberg, 2011) as opposed to SSA.

This factor was not specifically investigated for the inversions in this project, but we note it here in the interests of completeness. We do not believe that it would be a major cause of differences. Morlighem et al. (2013) concludes that the incomplete adjoint is an excellent approximation to the exact adjoint and does not have a significant effect on the convergence of inversions.

### Appendix C:    The unsuitability of diagnostic calculations for assessing the transferability of inversion products

A diagnostic model step calculates an instantaneous velocity from the given boundary conditions and geometry, without any time evolution. We ran diagnostic calculations in Úa using the fields of $B$ and $\beta^2$ produced by all three models as properties of the ice. We observe large discrepancies between the resulting velocity fields. The velocities produced when using the inversion outputs from ISSM and STREAMICE were significantly greater in magnitude than those calculated by their native models as part of the inversion process. Differences are seen in particular on the fastest flowing ice, and on the ice shelves, as seen in Fig. C1. The differences encountered indicate that transferring inversion products between models may not be a simple matter, since such large discrepancies appear in the velocities.

### C1    Differences at the grounding line

Investigating the phenomenon of large diagnostic velocity discrepancies further, we found that the position and definition of the grounding line is the major cause of the velocity differences. There are two reasons for inconsistencies to appear at the grounding line when transferring inversion products between models, which we shall discuss in this section.

Firstly, each model carries out inversions on a different mesh, and the outputs must be interpolated for use in other models. This is particularly important when transferring the $\beta^2$ fields, as the inversion output values of this parameter under the floating ice are not reliable. A remeshing and interpolation of data can result in some questionable values of $\beta^2$ at the grounding line, which can in turn significantly affect the calculated diagnostic velocity. In Fig. C1, the diagnostic velocities are calculated on Mesh2 (described in Appendix B3), which is coarser than the original mesh used for the inversions. Hence, the first panel

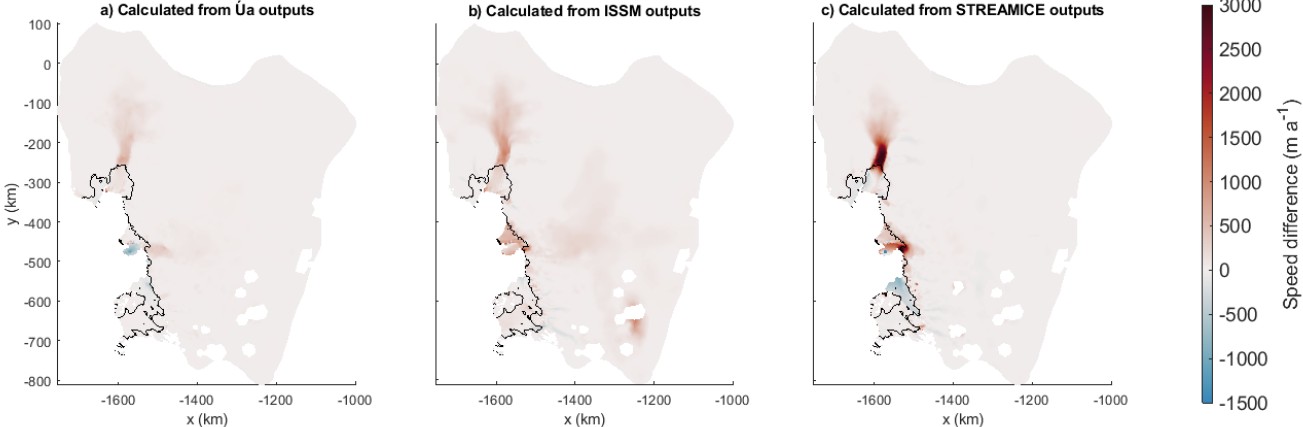

**Figure C1.** Differences between the speed calculated diagnostically in Úa using the $B$ and $\beta^2$ outputs from each model, and the original speed measurements used for the inversions. These calculations are performed on Mesh2, a coarser mesh than Úa's inversion.

demonstrates the effect of interpolating the $B$ and $\beta^2$ fields to a new mesh. The fact that the STREAMICE result displays the
635 largest velocity differences may be due to its original mesh being the most different from Mesh2.

Secondly, the models employ different treatments of the grounding line in their equations. Inversions in STREAMICE are calculated using a flotation relationship containing a Heaviside function which indicates whether ice in a mesh element is floating or grounded. Úa uses a modified version of this, as discontinuities in the equations can cause problems in the model's
numerical solvers. The Heaviside function is smoothed by use of a parameter named kH, which defines a length scale such that $\frac{1}{\text{kH}}$ is the height above flotation affected by the smoothing being applied. Thus, higher values of kH are closer to a true Heaviside function. ISSM employs the sub-element parameterisation scheme SEP2, as described in Seroussi et al. (2014). Some of the larger differences between the $\beta^2$ inversions in Fig. 5 (Sect. 4.3) can be seen at localised points along the grounding line, likely due to the differences in grounding line treatment.

For diagnostic calculations in Úa, we can change the value of kH to demonstrate the effects of varying grounding line regularisation. Figure C2 displays differences between diagnostic speeds calculated in Úa with different values of kH, and those calculated in the models from which the inversion outputs originate. An attempt can be made to replicate the diagnostic velocities of the original model as closely as possible within Úa's framework by 'tuning' the regularisation parameter, although this
is not likely to result in a good representation of the physics of the system.

For the ISSM outputs, lowering the value of kH causes the difference in calculated velocity to decrease on the ice shelf. However, there is a 'tipping point' beyond which decreasing the value further starts to increase the difference in the calculated velocity everywhere, especially on the grounded ice. For the results displayed in Fig. C1, we chose to set kH such that the

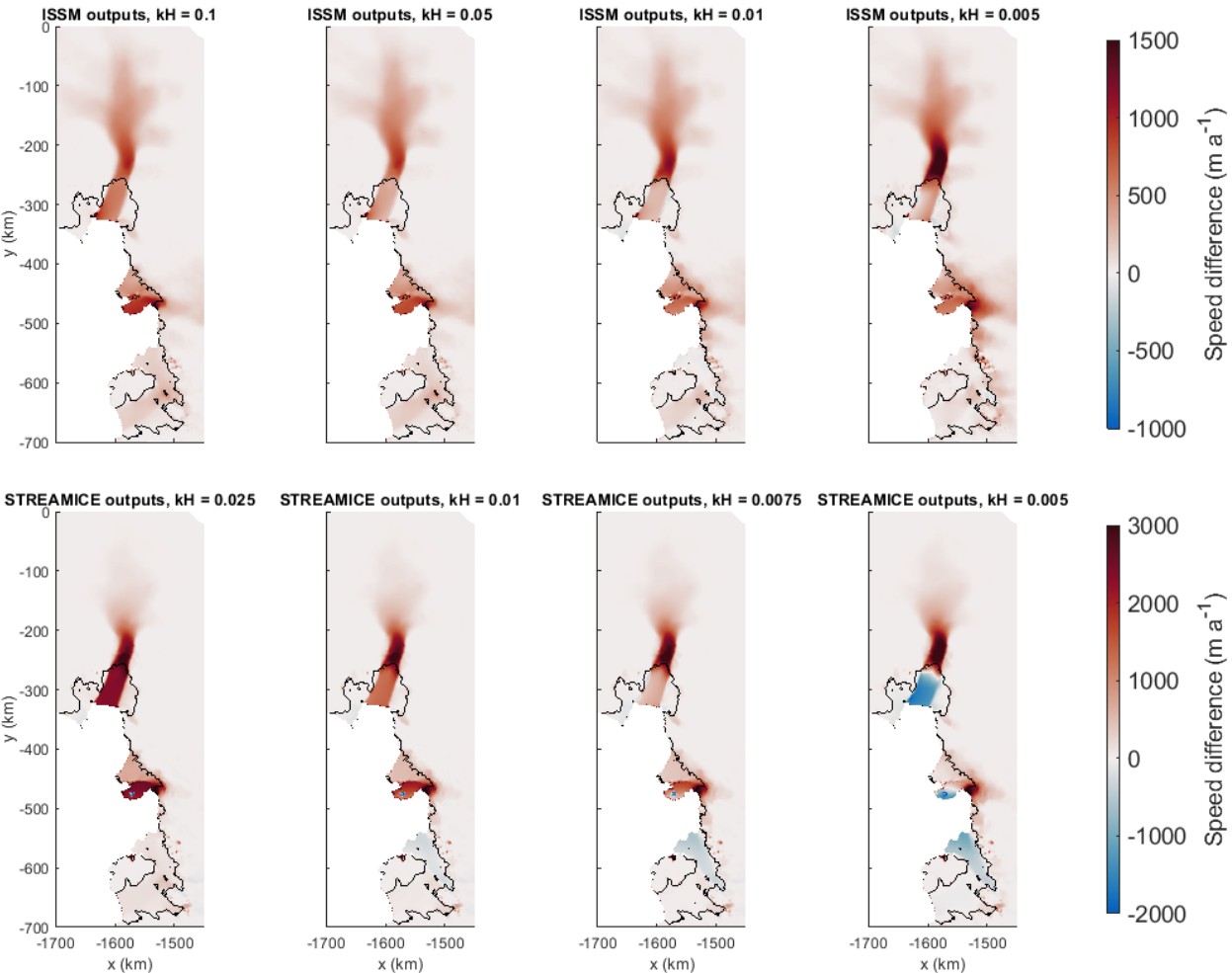

**Figure C2.** Differences between diagnostic speeds calculated in Úa using the ISSM and STREAMICE inversion outputs and a range of values for kH, and the speeds calculated in the original models. Note that each row uses a different colour scale.

difference on the ice shelf was minimised but this 'tipping point' was not passed, at $\mathtt{kH} = 0.02$.

The STREAMICE outputs follow a slightly different pattern. Beyond the 'tipping point', the difference on the ice shelves continues to follow the same trajectory, becoming negative as $\mathtt{kH}$ is lowered further. The change in the differences around this 'tipping point' is more sudden, and the difference on the grounded ice is greater in magnitude that that of the ISSM outputs.

Note that for these reasons, the bottom row of Fig. C2 uses a different colour scale and different values of $\mathtt{kH}$ than the top row. The value chosen for the results displayed in Fig. C1 was $\mathtt{kH} = 0.0065$.

## C2  Effects of grounding line differences on time-dependent simulations

The effect of the grounding line regularisation is different in time-dependent simulations. This is illustrated by the comparisons of speed and grounding line position shown in Fig. C3, which displays the results of three different simulations run in Úa, at

one year and ten years. The first is using Úa's own fields for $B$ and $\beta^2$, while the other two use the fields calculated by inversion in ISSM. The latter two are run using different values for $\mathtt{kH}$: one with $\mathtt{kH} = 0.02$, the value used for the diagnostic calculation in Fig. C1, and one with $\mathtt{kH} = 1$, the value used in the Úa simulation.

With both values for $\mathtt{kH}$, the ISSM fields produce a higher velocity than the fields from Úa. However, the difference is larger

when using the value which we chose for diagnostic calculations. We see that in the case where $\mathtt{kH} = 0.02$, the grounding line retreats further and the ice flows faster, despite this being the case which provided the closest velocities to the result using Úa's inversion products in the diagnostic calculation. This effect of using different values of $\mathtt{kH}$ is not entirely unexpected, as it is a regularisation parameter rather than a physical property of the ice. The definition of the grounding line has a large effect on basal melting, which is a physical process not included in the diagnostic calculation. This means that even though a

chosen value of $\mathtt{kH}$ can seem to produce similar results in a diagnostic setting, the additional physics of time-dependent flow can produce very different results.

This suggests that diagnostic calculations are not indicative of performance in time-dependent simulations, and that large velocity differences in the diagnostic calculations do not necessarily mean that similarly large differences will be present in

forward simulations. It also means that 'tuning' grounding line regularisation terms based on the diagnostics is not a method which should be used. Thus, for the time-dependent comparisons in Sect. 5, none of the models alter their treatment of the grounding line between experiments.

*Author contributions.*  All authors were involved in the conception of the project, and discussions throughout. JMB coordinated the project, and carried out the modelling work in Úa. TDdS and DG carried out the modelling work in ISSM and STREAMICE respectively. JMB led

the writing of the manuscript, and all authors provided comments and feedback during the editing process.

*Competing interests.* The authors declare that they have no competing interests.

*Acknowledgements.* The authors would like to thank Cyrille Mosbeux and one anonymous reviewer for their insightful feedback which helped to improve the manuscript, and Olivier Gagliardini for handling the editing of the paper.

This work is from the PROPHET project, a component of the International Thwaites Glacier Collaboration (ITGC). Support from National

Science Foundation (NSF: Grant #1739031) and Natural Environment Research Council (NERC: Grants NE/S006745/1, NE/S006796/1 and NE/T001607/1). ITGC Contribution No. 019.

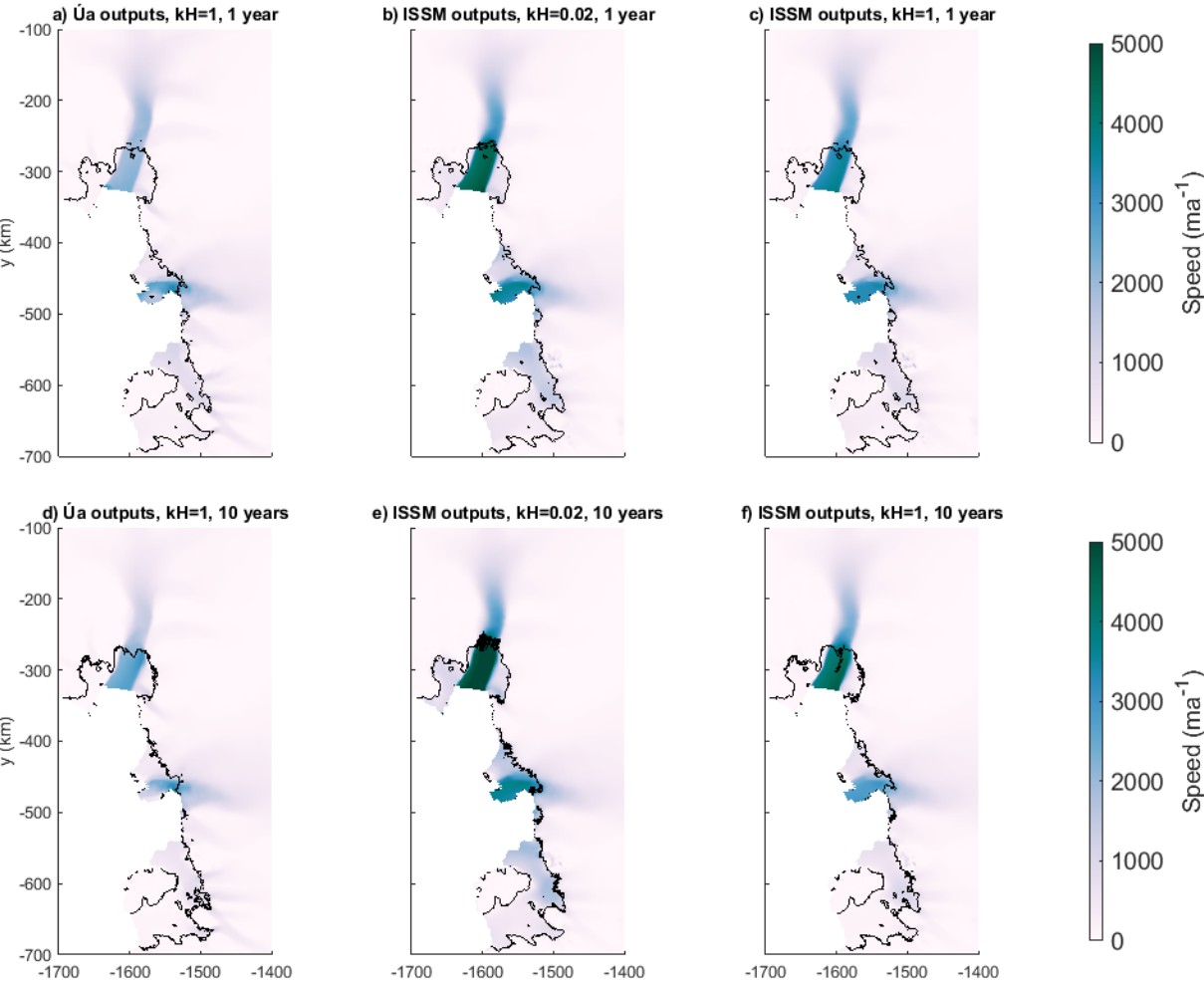

**Figure C3.** Speeds and grounding lines after 1 and 10 years of simulation in Úa using its own inversion outputs, and those of ISSM with two different values of `kH`.

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
