# Peer review of "The transferability of adjoint inversion products between different ice flow models"

_The Cryosphere, 2020_

## Referee Comment (RC1) · Anonymous Referee #1 · 21 Sep 2020

After reading the abstract, I was highly enthusiastic about this study, but felt it somewhat lacking after reading the entire paper.

Using rather vague language, the abstract suggests that in fact the inversions can transfer well, but in fact the results suggest something quite different. Simulations that after 40 years differ by a factor of 2 in predicted sea level rise can hardly be called successful, even if the bar has been set low by earlier studies. Likewise, diagnostic simulations that appear to differ by more than 2000 m/yr in places (see Figure 6).

The paper goes reasonably well through section 4.3, though I would argue that similarities and differences between the solutions are not that interesting. Even for a given model/inversion method, the correlation coefficient should be quite different for differing amounts of regularization (e.g., different members along the L-curve but with similar Printer-friendly version

misfits).

Once the paper gets in the diagnostic and prognostic simulations it becomes really not that helpful. What is shows is that one model, Ua, produces vastly different results with the different inversions, which may say more about Ua than about the generally transferability of these inversions. Is Ua highly sensitive to its tuning parameters? Hard to tell without seeing the other model results.

To really produce a robust finding, we need to see what the diagnostic simulations looks like with the other models (e.g. Fig 6 should have 9 panels, 3 models x 3 inversions). Same with the prognostic simulations (9 curves per figure). These are fairly low res simulations and not that difficult to run. Given that the paper includes co-authors from all 3 groups who have already done the work to setup the inversions, there is no reasons these additional simulations could not be run (I really feel this is essential for publication of this paper). It's in the authors best interest to do this, because the at present the spread in the results casts significant doubt on all 3 models, particularly Ua.

I could see if the models where of different order that results would be different, because the tuning process can compensate for some of the differences. But Ua and ISSM are both shallow shelf and both performed inversions on grids of similar complexity. The results should not be this different unless there are differences in the implementations of the same basic equations that really need to be elucidated. While some attribution is made to the implementation of the grounding line, these differences should not be that visible more than about 10 ice thicknesses inland, and as Figure B1 indicates, they extend more than 200-km inland. In figure 6 c, the really fast blobs in the interior seem to indicate some re-interpolation artifacts that should be fixed.

I would really like to see Ua turn off B for the grounded ice. I suspect a large part of the better fit is really a consequence of having twice the degrees of freedom at each point (B and Beta vs just Beta). I suspect some degree of over-fitting on the part of Ua
because the residual is better than the velocity errors (even if the formal errors indicate otherwise, they are not that good).

Specific Comments Line 235 - are these velocity (sqrt((u-uobs)2 + (v-vobs)2)) or speed (Vdiff) numbers. I would assume the former, but the context suggests the latter.

TCD

---

## Author Comment (AC1) · 28 Sep 2020

We thank the referee for their review, which will be helpful in improving our manuscript.

In response to the point about section 4, we would argue that similarities and differences between the inversion outputs are of great interest. The nature of inversion is that infinite solutions are possible, even when using the same inversion method, due to the ill-posedness of the problem. Our three models use different methods, and so there is theoretically no reason to assume that they would produce similar results. Through inversion processes, it could be possible to produce velocity fields with low misfit compared to velocity measurements, but with fields of B or Beta^2 which do not physically represent the system. This is why we feel it is important to examine and compare the

fields of B and Betaˆ2 before moving on to their application in transient simulations, and that the misfit alone is not necessarily an indicator of the quality of the inversions.

Regarding the transient simulations, a factor of 2 in the sea level contribution may sound large, but it is important to assess this within the right context. Several examples are listed in section 5.2, but this review makes it clear to us that significant expansion on this point, and some detail of other studies we are referencing, is required in the text to highlight the significance of our results. As one example, the control experiment in the initMIP-Antarctica comparison (Seroussi et al., 2019) shows a range of sea level contributions between -243mm and +167mm for the whole of Antarctica, with different models using different initialisation procedures. Within this range are a few examples of simulations run with the same models which differ by factors >3 due to differences in their initialisation. This, and results from other referenced studies, will be explicitly stated to aid in contextualising our results.

We agree with the referee that running diagnostic and transient simulations in all three models rather than just one is a good idea, and it is a point which the authors will act on. We hope that this will eliminate any doubt over the quality of the models we are using, reinforce our argument and help us to build a stronger case.

In general, the argument for Úa inverting for B across the entire domain is simply that we do not have definite information about the value of B. Not inverting for it assumes that there is zero uncertainty in the initial values imposed, which is not true. In the context of this work, we set out to compare the inversions produced by our models implementing their normal methods, and to show that despite the variety in the methods the results are physically robust. The difference in the B inversion is part of the variability between our methods, and not one of the controlled variables. An explanation to this effect will be added into section 2. The result of Úa turning off inversion for B is looked at in the appendix, section A2 (column c in Fig. A2). In this experiment, the speed misfit is higher, as the referee expects. The distribution of Betaˆ2 in this case remains consistent with the other experiments. This is mentioned in the main text (although

perhaps needs to be emphasised) in section 4.4.

In response to specific comments: There are indeed artifacts which need to be fixed in Fig. 6c. This will be done. Line 235 refers to the misfit (Vdiff), and the wording will be updated to clarify this.
* * *

---

## Referee Comment (RC2) · Cyrille Mosbeux (Referee) · 20 Oct 2020

Ice sheet model initialization is a crucial step to ensure that a model is as close as possible to the current state of the ice sheet. In this paper, the authors propose to infer two poorly known parameters (the basal friction and the stiffness or viscosity of the ice) in three different models and evaluate the differences between these models right after inversion and after a prognostic simulation, when transferring an initial state from one model to another.

The paper is relatively clear and I enjoyed reading it. The problem they set out to address is well-introduced with appropriate references. I found that the methodology and results were detailed well, although some sections and technical choices were

harder to follow due to back and forth between the main text and the appendices and sometimes lacking references to the appendices (see specific comments). The authors rightly recognize that the three models they use show significant differences (especially for the rate factor B) but that the large-scale distribution agrees well. While this is true for ISSM and STREAMICE, I am concerned by the results of Ua, which show particularly lower misfit between observed and model velocities. In this regard, the authors investigate various possibilities (in the Appendix) for explaining the variations between the models, but I would have liked to see more discussions on the L-Curve analysis. I detail this in a more specific comment but the difference in the misfit between the models brings up some questions about the regularization parameters used in the different models, e.g., are the three initial states really the minimum of each L-curve? Since an inversed state is particularly sensitive to the regularization parameters, it would be interesting to see the L-curve distribution and the location of the initial state picked for each model.

The paper then evaluates the transferability of Ua, ISSM and ICESTREAM initial state in Ua and with a coarser version of the Ua mesh. They conclude that this process is not straightforward and leads to substantial variations, but lies within the range of intercomparison experiments such as initMIP. However, in initMIP, the prognostic models are all different, which means that the differences are not only due to the initial state but also the physics of the transient models themselves (GL parametrization, etc.). Also, the models had various complexities while here, ISSM and Ua both use the SSA. In this regard, it is surprising that STREAMICE (L1L2) and ISSM (SSA) behave closer to each other than Ua (SSA) and ISSM. I think that the differences between the 3 prognostic simulations are relatively high and make it hard to believe that the transferability of initial states is a success. Comparing initial state transferability to the effect of different friction laws on sea level projections (Yu et al., 2018) is also a bit misleading since the latter involves changes in the physics of the model rather than difference in the numerical implementation (especially when Ua and ISSM both use the SSA). I would therefore recommend to temper these conclusions. In addition, I think that studying the

effect of the transferability from initial states to the other two models (Ua and ISSM to STREAMICE, Ua and STREAMICE to ISSM) could greatly benefit the study. This is a substantial effort (and the authors already did a significant number of sensitivity analysis) but it would provide a more comprehensive idea of the real transferability of initial states in the context of multi-model experiments like ISMIP6 (Serrousi et al., 2020), where one initial state could be provided to all the models.

Regardless of my concerns, the paper certainly deserves to be published (after revision) and will be useful to the community. The ability to use a similar initial state in different models for intercomparision experiments or to speed up some fastidious and repetitive initialization phases is of great interest to me. This paper shows the difficulty of the process and the remaining challenges we face in doing so.

You will find more specific comments in the attached document.

Please also note the supplement to this comment:
https://tc.copernicus.org/preprints/tc-2020-235/tc-2020-235-RC2-supplement.pdf

**Supplement:**

*Review of "The transferability of adjoint inversion products between different ice flow models" by Barnes et al. for The Cryosphere*

**General comments**

Ice sheet model initialization is a crucial step to ensure that a model is as close as possible to the current state of the ice sheet. In this paper, the authors propose to infer two poorly known parameters (the basal friction and the stiffness or viscosity of the ice) in three different models and evaluate the differences between these models right after inversion and after a prognostic simulation, when transferring an initial state from one model to another.

The paper is relatively clear and I enjoyed reading it. The problem they set out to address is well-introduced with appropriate references. I found that the methodology and results were detailed well, although some sections and technical choices were harder to follow due to back and forth between the main text and the appendices and sometimes lacking references to the appendices (see specific comments). The authors rightly recognize that the three models they use show significant differences (especially for the rate factor $B$) but that the large-scale distribution agrees well. While this is true for ISSM and STREAMICE, I am concerned by the results of Ua, which show particularly lower misfit between observed and model velocities. In this regard, the authors investigate various possibilities (in the Appendix) for explaining the variations between the models, but I would have liked to see more discussions on the L-Curve analysis. I detail this in a more specific comment but the difference in the misfit between the models brings up some questions about the regularization parameters used in the different models, e.g., are the three initial states really the minimum of each L-curve? Since an inversed state is particularly sensitive to the regularization parameters, it would be interesting to see the L-curve distribution and the location of the initial state picked for each model.

The paper then evaluates the transferability of Ua, ISSM and ICESTREAM initial state in Ua and with a coarser version of the Ua mesh. They conclude that this process is not straightforward and leads to substantial variations, but lies within the range of intercomparison experiments such as initMIP. However, in initMIP, the prognostic models are all different, which means that the differences are not only due to the initial state but also the physics of the transient models themselves (GL parametrization, etc.). Also, the models had various complexities while here, ISSM and Ua both use the SSA. In this regard, it is surprising that STREAMICE (L1L2) and ISSM (SSA) behave closer to each other than Ua (SSA) and ISSM. I think that the differences between the 3 prognostic simulations are relatively high and make it hard to believe that the transferability of initial states is a success. Comparing initial state transferability to the effect of different friction laws on sea level projections (Yu et al., 2018) is also a bit misleading since the latter involves changes in the physics of the model rather than difference in the numerical implementation (especially when Ua and ISSM both use the SSA). I would therefore recommend to temper these conclusions.

In addition, I think that studying the effect of the transferability from initial states to the other two models (Ua and ISSM to STREAMICE, Ua and STREAMICE to ISSM) could greatly benefit the study. This is a substantial effort (and the authors already did a significant number of sensitivity analysis) but it would provide a more comprehensive idea of the real transferability of initial states in the context of multi-model experiments like ISMIP6 (Serrousi et al., 2020), where one initial state could be provided to all the models.

Regardless of my concerns, the paper certainly deserves to be published (after revision) and will be useful to the community. The ability to use a similar initial state in different models for intercomparision experiments or to speed up some fastidious and repetitive initialization phases is of great interest to me. This paper shows the difficulty of the process and the remaining challenges we face in doing so.

**Specific comments**

- What are the boundary conditions for the different models? I assume that it could have some impact if they are different (especially the calving front). I guess that ISSM and Ua use very similar conditions, although the inside boundary in Ua is at the ice divide ($u$=0) while it is not for ISSM. Also, how is the calving front treated in STREAMICE? Some details about this could be included in the model description. I could only see a reference to the Dirichlet boundary condition at the ice divide ($u$=0) in *Sec. 5.1*. line 325.
- Section 4*:*
  - I am surprised by the relatively high-speed misfit of ISSM in Fig 3. Misfits exceeding 200 or 300 m/yr in most fast-flow regions seems very high. Especially when comparing with the other 2 models but also with other studies (e.g. Brondex et al. (2019), their Figure 4). Why is that so? One possible reason could be that in ISSM, since you also optimize the logarithm of the velocity misfit, you put a lot of weight on slow regions, limiting the optimization of fast flow regions. Regardless of the optimization, the SSA is also not particularly appropriate in these slow regions.
  - The problem you solve is ill-posed by nature, which is a common problem in glaciology but Ua seems particularly under-constrained here, giving a very nice (too nice?) velocity fit but creating a very different field of B (as mentioned by the authors in the Appendix A2).
  - For ISSM, is the inversion chosen here the real minimum of the L-curve? This minimum can be tricky to choose in a 3-D L-curve ($I$, $R_{p1}$ and $R_{p2}$).
  - The average misfit on the entire domain is also a bit misleading here since a large part of the domain is slow-flow regions where the absolute misfit will always be small, even with a poor inversion. Could you provide an average misfit for the fast-flow region (with a threshold of, for example, 50 or 100 m/a)?
- Please also specify the prior you use for the friction coefficient. I could not get any info on the friction prior before reading Appendix A4, which is, I think, never mentioned in the main text. Would it be useful to use an approximation based on the driving stress to construct the prior (instead of a constant value $\tau_b = 80$ kPa)?

- Check that all the Appendices are referenced in the main text.
- Appendix: algorithm performance (M1QN3 vs Interior Point) is one difference in the implementation of the optimization (for the gradient descent). However, another difference could be the way the adjoint model is derived in each model. Do all the methods consider a "self-adjoint" problem or do some of them use a complete gradient (see Martin and Monier, 2014)?
- Consider zooming on areas of interest like the grounding zone and the fast flow regions instead of always showing the entire domain. For example, it is very hard to compare the GL position of the different runs in Fig 8.

**Technical comments**

- Line 32: "[…] inverse problem may have an infinite number of arbitrarily different solutions […]": I would avoid using "arbitrarily" since the solution is still based on the method implementation (cost functions, regularization parameters, …).
- Line 33: I also think that "direct" problem (steady state or snapshot response of the model for the value of the inferred parameters) is better phrasing than "forward" (which has a transient connotation, as used later by the authors) since the forward response can be different for two initial states giving the same velocity misfit.
- Line 82: "inversion runs" instead of "inversions run".
- Line 98: consider developing the effective strain rate (or second invariant) term
- Line 118: consider using a vector $p = (p_1, p_2)$ here since it contains 2 parameters or components
- Line 130: "study" instead of "report"?
- Line 128: I think that the first term of the integral is a vector with $x$ and $y$ components and should use a norm like in Eq. (10), i.e. $\left\|\nabla\left(p_k - p_{k,prior}\right)\right\|$. How do you choose the prior values?
- Line 135: I am not sure that $b$ needs to be introduced in Eq. (7) since it is directly equaled to 1. Is $b$ always kept to 1 in different ISSM studies?
- Line 143: In Eq. (10), use $p_k$ instead of $p$ to keep consistency with Eq. (6)
- Line 159: $A_{i,j}$ is mentioned as the cell area but not used in the in Eqs. (11) and (12). Given that you invert for $p_1 = \sqrt{B}$, $B_0$ should be an initial estimate for $\sqrt{B}$ not $B$.
- Line 162: Are the regularization parameters also chosen with a L-curve analysis? Consider specifying it here too (since you did if for the two other models) or only mention once that you apply a L-curve analysis for the 3 models.
- Line 246: delete on of the closing parenthesis ")".
- Fig 4: first and second panels are both referred as (e), change for (d) and (e).
- Line 273: Is it only due to the fact that the regularization is conducted on the entire domain in Ua or to the prior used in Ua? This is answered in Appendix A4 but you do not refer to it in the main text.
- Line 282: Add "of" between "vicinity" and "the grounding line".

- Line 285: what do you mean by "to include peaks inside the rings of low values"? Do you directly constrain $\beta$ to stay positive during the inversion, like at each iteration?
- Line 335: The term "grounding line regularization" feels unclear until we read the appendix (making the reader jumping several pages). I think it is good to keep the details in the appendix but maybe a sentence to explain what "the grounding line regularization" is would be welcome in the main text. Also, the different values for the coefficient *kH* are given in the appendix but it is never explained what it refers to in the implementation of the grounding line dynamics (or position).
- Line 346: Did you test different interpolation methods (nearest neighbor vs linear)? Also, what append to $B$ and $\beta^2$ when directly inverting on Mesh2 and Mesh3? Are the values systematically higher than the interpolated fields, which could explain why the velocity is higher when interpolating?
- Fig. 6: Could you be more precise in the caption that this figure uses Mesh2 (in opposition to Fig. 3 using Mesh1)? Also, what is the reason for the very high misfit in panel (c) around x,y = (-1400, -600 km) ? Same for Fig B1.
- Fig. 7 and related text (line 360-368): I am confused here. Panels a, b and c display the same y-axis label (change in grounded are). Is that normal? If so, what is the difference between the panels? I understand from the text that 7c shows the change in grounded ice area but what about a and b? Is it also right that STREAMICE is ungrounding the most but that ISSM is losing more ice?
- Line 450: Missing "Pa" in the units for the friction parameter dipping below $10^2$ Pa m$^{-1/3}$ a$^{1/3}$
- Line 460: I agree with the authors' choice of capping the extreme values when calculating the correlation. Is this something you did for all the correlation values you got? I think it could be worth capping extremely low and high values. From my experience, $10^6$ Pa m$^{-1/3}$ a$^{1/3}$ is a good value for capping low friction coefficient but the threshold values below and above which the flow is virtually not affected could be tested in a more systematic way to see if it could increase the Pearson correlation coefficients.

**References**

Martin, N. and Monnier, J.: Adjoint accuracy for the full Stokes ice flow model: limits to the transmission of basal friction variability to the surface, The Cryosphere, 8, 721–741, https://doi.org/10.5194/tc-8-721-2014, 2014.

Seroussi, H., Nowicki, S., Simon, E., Abe-Ouchi, A., Albrecht, T., Brondex, J., Cornford, S., Dumas, C., Gillet-Chaulet, F., Goelzer, H., Golledge, N. R., Gregory, J. M., Greve, R., Hoffman, M. J., Humbert, A., Huybrechts, P., Kleiner, T., Larour, E., Leguy, G., Lipscomb, W. H., Lowry, D., Mengel, M., Morlighem, M., Pattyn, F., Payne, A. J., Pollard, D., Price, S. F., Quiquet, A., Reerink, T. J., Reese, R., Rodehacke, C. B., Schlegel, N.-J., Shepherd, A., Sun, S., Sutter, J., Van Breedam, J., van de Wal, R. S. W., Winkelmann, R., and Zhang, T.: initMIP-Antarctica: an ice sheet model initialization experiment of ISMIP6, The Cryosphere, 13, 1441–1471, https://doi.org/10.5194/tc-13-1441-2019, 2019.

Seroussi, H., Nowicki, S., Payne, A. J., Goelzer, H., Lipscomb, W. H., Abe-Ouchi, A., Agosta, C., Albrecht, T., Asay-Davis, X., Barthel, A., Calov, R., Cullather, R., Dumas, C., Galton-Fenzi, B. K., Gladstone, R., Golledge, N. R., Gregory, J. M., Greve, R., Hattermann, T., Hoffman, M. J., Humbert, A., Huybrechts, P., Jourdain, N. C., Kleiner, T., Larour, E., Leguy, G. R., Lowry, D. P., Little, C. M., Morlighem, M., Pattyn, F., Pelle, T., Price, S. F., Quiquet, A., Reese, R., Schlegel, N.-J., Shepherd, A., Simon, E., Smith, R. S., Straneo, F., Sun, S., Trusel, L. D., Van Breedam, J., van de Wal, R. S. W., Winkelmann, R., Zhao, C., Zhang, T., and Zwinger, T.: ISMIP6 Antarctica: a multi-model ensemble of the Antarctic ice sheet evolution over the 21st century, The Cryosphere, 14, 3033–3070, https://doi.org/10.5194/tc-14-3033-2020, 2020.

---

## Author Comment (AC2) · 11 Dec 2020

We thank Cyrille Mosbeux for the detailed and constructive review, which will help us to improve our manuscript. Responses to specific and technical comments are included in the supplement.

To address the general comments in the review, and those of Referee #1, changes are being made to the manuscript as follows:

- Addition of more references to relevant sections of the appendix, along with summaries of the information contained within where appropriate, to increase clarity in the main body of the text.

- Addition of a new section to the appendix, giving more detail on the choice of regularisation and displaying relevant L-curves.

- Addition of experiments in which all inversion products are used in ISSM and STREAMICE for time-dependent simulations. These experiments have already been run, and the results are available to be added to the manuscript.

- Expansion of the discussion to give more detail on comparisons being made to other studies, and the limitations of these comparisons, leading to a tempering of conclusions regarding the success of transferability.

- Editing of the abstract and introduction to reflect the revised tone of the conclusions, and temper the expectations of the reader.

- Merge section 5.1 into Appendix B, and summarise in the main text with clear reference to the appendix. We conclude that the diagnostic results are not a good indicator of time-dependent performance, which is worthy of note thus should be kept in an appendix, but this section in its current form seems to confuse the narrative and take some attention away from the main results (particularly based on the comments of Referee #1).

Please also note the supplement to this comment:
https://tc.copernicus.org/preprints/tc-2020-235/tc-2020-235-AC2-supplement.pdf

**Supplement:**

We thank Cyrille Mosbeux for the detailed and constructive review, which will help us to improve our manuscript. Comments from the review are displayed in blue.

To address the general comments in the review, and those of Referee #1, actions are being taken (in fact, much has been done already) as follows:

- Addition of more references to relevant sections of the appendix, along with summaries of the information contained within where appropriate, to increase clarity in the main body of the text.
- Addition of a new section to the appendix, giving more detail on the choice of regularisation and displaying relevant L-curves.
- Addition of experiments in which all inversion products are used in ISSM and STREAMICE for time-dependent simulations. These experiments have already been run, and the results are available to be added to the manuscript.
- Expansion of the discussion to give more detail on comparisons being made to other studies, and the limitations of these comparisons, leading to a tempering of conclusions regarding the success of transferability.
- Editing of the abstract and introduction to reflect the revised tone of the conclusions, and temper the expectations of the reader.
- Merge section 5.1 into Appendix B, and summarise in the main text with clear reference to the appendix. We conclude that the diagnostic results are not a good indicator of time-dependent performance, which is worthy of note thus should be kept in an appendix, but this section in its current form seems to confuse the narrative and take some attention away from the main results (particularly based on the comments of Referee #1).

**Specific comments**

- What are the boundary conditions for the different models? I assume that it could have some impact if they are different (especially the calving front). I guess that ISSM and Ua use very similar conditions, although the inside boundary in Ua is at the ice divide ($u$=0) while it is not for ISSM. Also, how is the calving front treated in STREAMICE? Some details about this could be included in the model description. I could only see a reference to the Dirichlet boundary condition at the ice divide ($u$=0) in *Sec. 5.1*. line 325.

ISSM has a Dirichlet boundary condition at the edge of the domain on grounded ice, given values based on the observed velocities. In STREAMICE, the grounded edges are given a no-flow boundary condition. These boundaries are sufficiently far from the area of interest to make no difference to the outcome.

All models apply an ice front stress condition. In Úa this is at the domain boundary, while in ISSM and STREAMICE the location of the ice front is fixed by an ice/ocean mask created using the geometry data.

This information has been added to section 3.2.

- Section 4:
  - I am surprised by the relatively high-speed misfit of ISSM in Fig 3. Misfits exceeding 200 or 300 m/yr in most fast-flow regions seems very high. Especially when comparing with the other 2 models but also with other studies (e.g. Brondex et al. (2019), their Figure 4). Why is that so? One possible reason could be that in ISSM, since you also optimize the logarithm of the velocity misfit, you put a lot of weight on slow regions, limiting the optimization of fast flow regions. Regardless of the optimization, the SSA is also not particularly appropriate in these slow regions.

It is true that ISSM produces higher misfit than the other two models in our study, and this is likely down to the equal weighting given to the absolute and logarithmic misfits. Perhaps a detailed study of these weighting choices could be the subject of some future work.

However, we do not believe the misfit is unreasonably high. There are not many areas which produce misfit >200 m/a. Regarding the comparison to Brondex et al. (2019), it appears that our misfit falls somewhere in the middle of the range of inferred states examined in that paper, similar to their Fig. 4(b).

  - The problem you solve is ill-posed by nature, which is a common problem in glaciology but Ua seems particularly under-constrained here, giving a very nice (too nice?) velocity fit but creating a very different field of B (as mentioned by the authors in the Appendix A2).

A large amount of the difference seen in values for B, and the misfits, is a direct result of Úa inverting for B over the entire domain, whereas the other models do not. This is common practice in Úa and this project, while controlling certain input datasets, set out to compare inversions carried out under normal working practices in each model.

Relevant to both this and the point above, in Figure A2, column c, we display the result of carrying out an inversion in Úa following the practice of ISSM as closely as possible, and obtain a similar misfit to the ISSM inversion. In the revised manuscript, this will be referred to in the main text when discussing the misfit of our inversions.

  - For ISSM, is the inversion chosen here the real minimum of the L-curve? This minimum can be tricky to choose in a 3-D L-curve ($I, Rp1$ and $Rp2$).

In the case of ISSM, the inversions for B and beta^2 are carried out separately, one after the other, as described in subsection 2.2.2. This means that the two regularisation parameters can be chosen independently. This results in two separate 2D L-curves, rather than one in 3D. The same cannot be said for the other models, however. We are adding a new appendix section to explain the regularisation choices and display the L-curves. The issue of multiple regularisation parameters and how these were approached in each model will be discussed here. The appendix section will be referred to in section 2.2.

  - The average misfit on the entire domain is also a bit misleading here since a large part of the domain is slow-flow regions where the absolute misfit will always be small, even with a poor inversion. Could

> you provide an average misfit for the fast-flow region (with a threshold of, for example, 50 or 100 m/a)?

This is a good point, and average misfits for a defined velocity threshold will be added to the analysis in section 4.1.

- Please also specify the prior you use for the friction coefficient. I could not get any info on the friction prior before reading Appendix A4, which is, I think, never mentioned in the main text. Would it be useful to use an approximation based on the driving stress to construct the prior (instead of a constant value $\tau b$=80 kPa)?

The priors will be specified in section 2.2, and references to appendices will be improved throughout the manuscript. The use of priors defined in different ways, including one based on driving stress, could be an interesting topic for further experimentation, but beyond the intended scope of this project. For the purposes of this paper we chose one spatially varying prior to contrast with a spatially uniform prior and the original priors used by each model.

- Check that all the Appendices are referenced in the main text.

This is being improved upon in revisions to the manuscript.

- Appendix: algorithm performance (M1QN3 vs Interior Point) is one difference in the implementation of the optimization (for the gradient descent). However, another difference could be the way the adjoint model is derived in each model. Do all the methods consider a "self-adjoint" problem or do some of them use a complete gradient (see Martin and Monier, 2014)?

The models do use slightly different derivations of the adjoint. ISSM uses the exact adjoint described in Morlighem et al. (2013). Following that paper, we prefer to refer to the alternative as "incomplete" rather than "self-adjoint". STREAMICE uses the method described in Goldberg et al. (2016), but with a relatively weak tolerance placing it somewhere between the exact and incomplete adjoints.

We did not consider these among the differences in Appendix A, and will mention this as a possible factor alongside the others in the revised manuscript.

- Consider zooming on areas of interest like the grounding zone and the fast flow regions instead of always showing the entire domain. For example, it is very hard to compare the GL position of the different runs in Fig 8.

A good point! We will zoom in to display the areas of interest in Fig. 8 and make the differences much clearer to see.

**Technical comments**

Thank you for pointing out typographical errors, and areas which require greater clarity. These shall all be addressed. A few of these technical comments require specific responses.

- Line 162: Are the regularization parameters also chosen with a L-curve analysis? Consider specifying it here too (since you did if for the two other models) or only mention once that you apply a L-curve analysis for the 3 models.

The parameters for STREAMICE were selected based on an L-curve analysis from previous experiments run in the model (Goldberg et al., 2019), rather than independently for this work. This will be made clear and the reference cited in a new appendix section discussing regularisation choices.

- Line 285: what do you mean by "to include peaks inside the rings of low values"? Do you directly constrain $\beta$ to stay positive during the inversion, like at each iteration?

There was not a constraint within the inversion. The effect is a result of post-processing of the data. This has been clarified in the text.

- Line 335: The term "grounding line regularization" feels unclear until we read the appendix (making the reader jumping several pages). I think it is good to keep the details in the appendix but maybe a sentence to explain what "the grounding line regularization" is would be welcome in the main text. Also, the different values for the coefficient $kH$ are given in the appendix but it is never explained what it refers to in the implementation of the grounding line dynamics (or position).

Clarity on this matter is being improved by the merging of section 5.1 and Appendix B. Improvements are being made throughout on references to the appendices, and reducing the need to jump back and forth between pages.

- Fig. 7 and related text (line 360-368): I am confused here. Panels a, b and c display the same y-axis label (change in grounded are). Is that normal? If so, what is the difference between the panels? I understand from the text that 7c shows the change in grounded ice area but what about a and b? Is it also right that STREAMICE is ungrounding the most but that ISSM is losing more ice?

The repeated labels are an image processing error which will be corrected. Panel a) should read "Loss of volume above flotation (Gt)" and panel b) should read "Change in total ice mass (Gt)".

This entire section will be expanded to include results from the other models, and to improve clarity.

- Line 460: I agree with the authors' choice of capping the extreme values when calculating the correlation. Is this something you did for all the correlation values you got? I think it could be worth capping extremely low and high values. From my experience, $10^6$ Pa m$^{-1/3}$ a$^{1/3}$ is a good value for capping low friction coefficient but the threshold values below and above which the flow is virtually not affected could be tested in a more systematic way to see if it could increase the Pearson correlation coefficients.

Several thresholds were tried for capping extreme values, and the chosen values are those which were deemed to strike the best balance between preserving the shape of the data and discarding anomalous spikes. This was not done for every correlation value, but only where necessary in some of the comparisons presented in Table A1. The affected values are indicated in the table and caption.

**References**

Brondex, J., Gillet-Chaulet, F. and Gagliardini, O., 2019. Sensitivity of centennial mass loss projections of the Amundsen basin to the friction law. *The Cryosphere*, *13*(1), pp.177-195.

Morlighem, M., Seroussi, H., Larour, E. and Rignot, E., 2013. Inversion of basal friction in Antarctica using exact and incomplete adjoints of a higher-order model. *Journal of Geophysical Research: Earth Surface*, *118*(3), pp.1746-1753.

Goldberg, D., Narayanan, S.H.K., Hascoet, L. and Utke, J., 2016. An optimized treatment for algorithmic differentiation of an important glaciological fixed-point problem.

Goldberg, D.N., Gourmelen, N., Kimura, S., Millan, R. and Snow, K., 2019. How accurately should we model ice shelf melt rates?. *Geophysical Research Letters*, *46*(1), pp.189-199.

---

## Referee Report (RR1)

**Second round of review of "The transferability of adjoint inversion products between different ice flow models" by Barnes et al. for The Cryosphere**

*I really enjoyed the paper and the addition made by the authors in this revised version. My general and specific comments are displayed in italic green text here after.*

A separate document showing all changes to the manuscript is attached to this submission.

Earlier responses to the reviews are copied below for convenience.

In summary, to address the comments of the reviewers, revisions have been made as follows:

- Addition of experiments in which all inversion products are used in ISSM and STREAMICE for time-dependent simulations, and the necessary rewriting of section 5 along with new figures 6 and 7. Please note, all experiments presented in section 5 are new, including the Úa simulations. The revision process and comparison between forward runs offered us an opportunity to identify an issue which was causing results from Úa to differ significantly more than they should. The issue was due to the values of beta^2 on the ungrounded ice causing areas of regrounding after an initial advance of the grounding line. This was rectified by setting the beta^2 values in this region manually, and is discussed in the text. Thus, the factor of 2 in the differences from the original experiments is now greatly reduced.
- Addition of some numerical values to the abstract, to provide realistic expectations to the reader about the outcomes of the paper.

*Thank you for this addition. If space is not a problem, I would also add the value in mm in addition to the percentage variability of sea level.*

- Expansion of the discussion to give more detail on comparisons being made to other studies, and pointing out the limitations of some comparisons.
- Addition of more references to relevant sections of the appendix, along with summaries of the information contained within where appropriate, to increase clarity in the main body of the text.
- Addition of a new section to the appendix (now Appendix A), giving more detail on the choice of regularisation and displaying relevant L-curves.
- Addition of a new section to the appendix (now Appendix B5) addressing differences in the derivation of the adjoint.
- What was formerly section 5.1 has been merged into Appendix C, and referenced in the main text. The conclusion that the diagnostic results are not a good indicator of time-dependent performance is worthy of note in an appendix, but the details of this were confusing the narrative in their previous position.
- Information on the choice of priors added to section 3.1.
- Information on boundary conditions added to section 3.2.

- Average misfits for some defined velocity thresholds added to the analysis in section 4.1.

**Response to RC1**

We thank the referee for their review, which will be helpful in improving our manuscript.

In response to the point about section 4, we would argue that similarities and differences between the inversion outputs are of great interest. The nature of inversion is that infinite solutions are possible, even when using the same inversion method, due to the ill-posedness of the problem. Our three models use different methods, and so there is theoretically no reason to assume that they would produce similar results. Through inversion processes, it could be possible to produce velocity fields with low misfit compared to velocity measurements, but with fields of B or Beta^2 which do not physically represent the system. This is why we feel it is important to examine and compare the fields of B and Beta^2 before moving on to their application in transient simulations, and that the misfit alone is not necessarily an indicator of the quality of the inversions.

Regarding the transient simulations, a factor of 2 in the sea level contribution may sound large, but it is important to assess this within the right context. Several examples are listed in section 5.2, but this review makes it clear to us that significant expansion on this point, and some detail of other studies we are referencing, is required in the text to highlight the significance of our results. As one example, the control experiment in the initMIP-Antarctica comparison (Seroussi et al., 2019) shows a range of sea level contributions between -243mm and +167mm for the whole of Antarctica, with different models using different initialisation procedures. Within this range are a few examples of simulations run with the same models which differ by factors >3 due to differences in their initialisation. This, and results from other referenced studies, will be explicitly stated to aid in contextualising our results.

We agree with the referee that running diagnostic and transient simulations in all three models rather than just one is a good idea, and it is a point which the authors will act on. We hope that this will eliminate any doubt over the quality of the models we are using, reinforce our argument and help us to build a stronger case.

In general, the argument for Úa inverting for B across the entire domain is simply that we do not have definite information about the value of B. Not inverting for it assumes that there is zero uncertainty in the initial values imposed, which is not true. In the context of this work, we set out to compare the inversions produced by our models implementing their normal methods, and to show that despite the variety in the methods the results are physically robust. The difference in the B inversion is part of the variability between our methods, and not one of the controlled variables. An explanation to this effect will be added into section 2. The result of Úa turning off inversion for B is looked at in the appendix, section A2 (column c in Fig. A2). In this experiment, the speed misfit is higher, as the referee expects. The distribution of Beta^2 in

this case remains consistent with the other experiments. This is mentioned in the main text (although perhaps needs to be emphasised) in section 4.4.

In response to specific comments:
There are indeed artifacts which need to be fixed in Fig. 6c. This will be done.
Line 235 refers to the misfit (Vdiff), and the wording will be updated to clarify this.

**Response to RC2**

We thank Cyrille Mosbeux for the detailed and constructive review, which will help us to improve our manuscript. Comments from the review are displayed in blue.

To address the general comments in the review, and those of Referee #1, actions are being taken (in fact, much has been done already) as follows:

- Addition of more references to relevant sections of the appendix, along with summaries of the information contained within where appropriate, to increase clarity in the main body of the text.
- Addition of a new section to the appendix, giving more detail on the choice of regularisation and displaying relevant L-curves.
- Addition of experiments in which all inversion products are used in ISSM and STREAMICE for time-dependent simulations. These experiments have already been run, and the results are available to be added to the manuscript.
- Expansion of the discussion to give more detail on comparisons being made to other studies, and the limitations of these comparisons, leading to a tempering of conclusions regarding the success of transferability.
- Editing of the abstract and introduction to reflect the revised tone of the conclusions, and temper the expectations of the reader.
- Merge section 5.1 into Appendix B, and summarise in the main text with clear reference to the appendix. We conclude that the diagnostic results are not a good indicator of time-dependent performance, which is worthy of note thus should be kept in an appendix, but this section in its current form seems to confuse the narrative and take some attention away from the main results (particularly based on the comments of Referee #1).

*Thank you to the authors for the work on this revision and their great attention to details. The authors have clearly accounted for my previous comments, running a significant amount of new simulations and clearly detailing modelling choices and inherent limitations of each model. I think this extra work has strengthened the manuscript and gives the reader more transparency on what to expect when transferring inversion products from one model to another. I have only a few additional comments (see green italic text), which are suggestions rather than critical remarks.*

*In the end, I am very happy with this version of the manuscript. Like the authors, I particularly think that the results of the forward modeling over a 40-year period exhibiting ~10 to ~30-40%*

difference in SLR contribution is very encouraging for the application of inversion product transfer in the future. As underlined by the authors, I am sure that this work will be useful to many other modelers interested in such transfer of inversion product, either to save computing time (by reusing outputs from previous studies) or for intercomparison experiments.

- *Line 24: specify the location of Ice Stream E (e.g., tributary of Ross Ice Shelf, Antarctica).*
- *Line 29-32: add a reference to Gillet-Chaulet (2020) who shows a promising ensemble Kalman filter method.*
- *Line 74: I would specify the period of observation: "[...] and has been accelerating over the last decades (Sutterley et al., 2014)"*
- *Line 75: I would change "can produce" for "produce"*
- *Line 92 to 97:*
  - *Although the paper is really aimed at ice-sheet modelers that know the different ice-flow equations, I would give a small explanation about what is L1L2. Maybe one sentence stating the hypothesis and differences between SSA and L1L2.*
  - *I would also specify here that ISSM and Ua are both FE models (since you introduce STREAMICE as "not a purely FE model" in line 157). Also indicate why STREAMICE is not a purely FE model.*
- *Line 105: I think "rate factors are [...]" should be singular*
- *Line 179: "inversion processes have performed" instead of "[...] has performed"*
- *Line 226: change "inversions run" run for "inversion runs" or just "inversions" (same in the caption of Figure B2)*
- *Line 246-248 and Appendix B: It is interesting to see that Ua misfit largely increases when using B from ISSM and only inverts for $\beta^2$ (Figure 3b and B2c look really alike, especially when looking at the misfit on the Thwaites glacier tongue). I think you rightly point out that, in addition to the logarithmic cost function used in ISSM, the fact that the inversion of both parameters is sequent (i.e., B and then $\beta^2$ ) and not simultaneous could have a large impact.*
- *Line 294: I think that this statement could be linked with my previous comment: $\beta^2$ is fairly similar in all the models but the inversion of B in ISSM seems to suffer from the fact that $\beta^2$ is only using its prior value when inverting for B. Did you try to invert for $\beta^2$ first and then B, or even reinverting for B a second time (I detail this a bit further)? Notice that I understand STREAMICE and ISSM showing similar B values on the shelve maybe suggest that this is not that important.*
- *Line 326: Should not it be "sequential nature of ISSM" instead of "non-sequential" (since ISSM uses a sequence of B and $\beta^2$ inversion)?*
- *Line 375: what do you expect or see as differences between the GL retreat of SSA models and the L1L2? I think that this should be slightly develop since you point out the difference in the equations.*
- *Line 424: "They each use different methods and employ different techniques during the inversion process, [...]" looks a bit repetitive, maybe change for "They each use different inversion techniques [...]"?*
- *Line 463: add a comma between "[...] are selected" and "the corresponding [...]"*

- *Line 492-493: please consider adding the mean magnitude misfit obtained with the Interior Point algorithm (for direct comparison with the value you give for M1QN3).*
- *Line 516: "to be" instead of "be be"*
- *Line 519: remind here the value of the Pearson correlation so that the reader can directly make the comparison with the value 0.513 given in line 522.*
- *Figure B3: Why is ISSM $\beta^2$ so low over Pine Island fast ice when using Priors 1? Maybe add a potential explanation?*
- *Please see a few additional remarks in the "specific comments" section below.*

**Specific comments**

• What are the boundary conditions for the different models? I assume that it could have some impact if they are different (especially the calving front). I guess that ISSM and Ua use very similar conditions, although the inside boundary in Ua is at the ice divide (*u*=0) while it is not for ISSM. Also, how is the calving front treated in STREAMICE? Some details about this could be included in the model description. I could only see a reference to the Dirichlet boundary condition at the ice divide (*u*=0) in *Sec. 5.1*. line 325.

ISSM has a Dirichlet boundary condition at the edge of the domain on grounded ice, given values based on the observed velocities. In STREAMICE, the grounded edges are given a no-flow boundary condition. These boundaries are sufficiently far from the area of interest to make no difference to the outcome. All models apply an ice front stress condition. In Úa this is at the domain boundary, while in ISSM and STREAMICE the location of the ice front is fixed by an ice/ocean mask created using the geometry data.
This information has been added to section 3.2.

• Section 4:
o I am surprised by the relatively high-speed misfit of ISSM in Fig 3.

Misfits exceeding 200 or 300 m/yr in most fast-flow regions seems very high. Especially when comparing with the other 2 models but also with other studies (e.g. Brondex et al. (2019), their Figure 4). Why is that so? One possible reason could be that in ISSM, since you also optimize the logarithm of the velocity misfit, you put a lot of weight on slow regions, limiting the optimization of fast flow regions. Regardless of the optimization, the SSA is also not particularly appropriate in these slow regions.

It is true that ISSM produces higher misfit than the other two models in our study, and this is likely down to the equal weighting given to the absolute and logarithmic misfits. Perhaps a detailed study of these weighting choices could be the subject of some future work.

However, we do not believe the misfit is unreasonably high. There are not many areas which produce misfit >200 m/a. Regarding the comparison to Brondex et al. (2019), it appears that our misfit falls somewhere in the middle of the range of inferred states examined in that paper, similar to their Fig. 4(b).

*Thank you for this extra comment. I agree that the misfit of ISSM is similar to Figure 4b in Brondex et al. (2019). However, this high misfit is due to their low weighting of the gradients with respect to the viscosity parameter. When they increase the weighting (having, I believe a ratio 1:1) they get Fig 4c, which really reduces the misfit.*

*However, I understand that the effect of linear vs logarithmic misfit will increase the number of simulations shown in this present study and that the authors effort in the number of simulations is already impressive.*

*One last note, as it is highlighted in the paper, the "bad" result of the steady misfit in ISSM is not necessarily a problem when looking at the evolution of the misfit after a forward simulation. Maybe it is worth to clearly state this (maybe in the discussion) ; it is somewhat understood when reading the paper but emphasizing this could be useful.*

o   The problem you solve is ill-posed by nature, which is a common problem in glaciology but Ua seems particularly under-constrained here, giving a very nice (too nice?) velocity fit but creating a very different field of B (as mentioned by the authors in the Appendix A2).

A large amount of the difference seen in values for B, and the misfits, is a direct result of Úa inverting for B over the entire domain, whereas the other models do not. This is common practice in Úa and this project, while controlling certain input datasets, set out to compare inversions carried out under normal working practices in each model.

Relevant to both this and the point above, in Figure A2, column c, we display the result of carrying out an inversion in Úa following the practice of ISSM as closely as possible, and obtain a similar misfit to the ISSM inversion. In the revised manuscript, this will be referred to in the main text when discussing the misfit of our inversions.

o   For ISSM, is the inversion chosen here the real minimum of the L- curve? This minimum can be tricky to choose in a 3-D L-curve ($I$,$Rp1$ and $Rp2$).

In the case of ISSM, the inversions for B and beta^2 are carried out separately, one after the other, as described in subsection 2.2.2. This means that the two regularisation parameters can be chosen independently. This results in two separate 2D L-curves, rather than one in 3D. The same cannot be said for the other models, however. We are adding a new appendix section to explain the regularisation choices and display the L-curves. The issue of multiple regularisation parameters and how these were approached in each model will be discussed here. The appendix section will be referred to in section 2.2.

*Thank you for the addition of the L-curve analysis. It really eases the reading in the regularization choices of each model.*

*The effect of the logarithmic velocity misfit in ISSM has been mentioned earlier. You also mention that the separate (sequential) inversion of both parameters could also be a cause for*

*the higher misfit with respect to the two other models. I think it is a very good point. Maybe this is not a common practice in ISSM but what about a looping algorithm allowing to reinvert B a second time after the two first inversions B and $\beta^2$ :*

$$B_0 \rightarrow \beta_0^2 \rightarrow B_1 \rightarrow \beta_1^2 \rightarrow B_2 \rightarrow \dots$$

*Until the misfit converges to a minimum value. This could help reaching a lower misfit value (by "relaxing" potential local minima)? Indeed, after each inversion you will have a better representation of each parameter (e.g. $\beta^2$) which might help a new inversion of the previous parameter (e.g. B).*

o The average misfit on the entire domain is also a bit misleading here since a large part of the domain is slow-flow regions where the absolute misfit will always be small, even with a poor inversion. Could you provide an average misfit for the fast-flow region (with a threshold of, for example, 50 or 100 m/a)?

This is a good point, and average misfits for a defined velocity threshold will be added to the analysis in section 4.1.

• Please also specify the prior you use for the friction coefficient. I could not get any info on the friction prior before reading Appendix A4, which is, I think, never mentioned in the main text. Would it be useful to use an approximation based on the driving stress to construct the prior (instead of a constant value $\tau b$=80 kPa)?

The priors will be specified in section 2.2, and references to appendices will be improved throughout the manuscript. The use of priors defined in different ways, including one based on driving stress, could be an interesting topic for further experimentation, but beyond the intended scope of this project. For the purposes of this paper we chose one spatially varying prior to contrast with a spatially uniform prior and the original priors used by each model.

• Check that all the Appendices are referenced in the main text.

This is being improved upon in revisions to the manuscript.

• Appendix: algorithm performance (M1QN3 vs Interior Point) is one difference in the implementation of the optimization (for the gradient descent). However, another difference could be the way the adjoint model is derived in each model. Do all the methods consider a "self-adjoint" problem or do some of them use a complete gradient (see Martin and Monier, 2014)?

The models do use slightly different derivations of the adjoint. ISSM uses the exact adjoint described in Morlighem et al. (2013). Following that paper, we prefer to refer to the alternative as "incomplete" rather than "self-adjoint". STREAMICE uses the method described in Goldberg

et al. (2016), but with a relatively weak tolerance placing it somewhere between the exact and incomplete adjoints.

We did not consider these among the differences in Appendix A, and will mention this as a possible factor alongside the others in the revised manuscript.

• Consider zooming on areas of interest like the grounding zone and the fast flow regions instead of always showing the entire domain. For example, it is very hard to compare the GL position of the different runs in Fig 8.

A good point! We will zoom in to display the areas of interest in Fig. 8 and make the differences much clearer to see.

*Thank you. The figure 7 in the new version is indeed much more legible. Please, indicate that the black line is the grounding line. Could you also plot the initial position of the grounding line (right after inversion)? It would allow a better visualization of the grounding line migration.*

**Technical comments**

Thank you for pointing out typographical errors, and areas which require greater clarity. These shall all be addressed. A few of these technical comments require specific responses.

• Line 162: Are the regularization parameters also chosen with a L-curve analysis? Consider specifying it here too (since you did if for the two other models) or only mention once that you apply a L-curve analysis for the 3 models.

The parameters for STREAMICE were selected based on an L-curve analysis from previous experiments run in the model (Goldberg et al., 2019), rather than independently for this work. This will be made clear and the reference cited in a new appendix section discussing regularisation choices.

• Line 285: what do you mean by "to include peaks inside the rings of low values"? Do you directly constrain $\beta$ to stay positive during the inversion, like at each iteration?

There was not a constraint within the inversion. The effect is a result of post- processing of the data. This has been clarified in the text.

• Line 335: The term "grounding line regularization" feels unclear until we read the appendix (making the reader jumping several pages). I think it is good to keep the details in the appendix but maybe a sentence to explain what "the grounding line regularization" is would be welcome in the main text. Also, the different values for the coefficient *kH* are given in the appendix but it is never explained what it refers to in the implementation of the grounding line dynamics (or position).

Clarity on this matter is being improved by the merging of section 5.1 and Appendix B. Improvements are being made throughout on references to the appendices, and reducing the need to jump back and forth between pages.

• Fig. 7 and related text (line 360-368): I am confused here. Panels a, b and c display the same y-axis label (change in grounded are). Is that normal? If so, what is the difference between the panels? I understand from the text that 7c shows the change in grounded ice area but what about a and b? Is it also right that STREAMICE is ungrounding the most but that ISSM is losing more ice?

The repeated labels are an image processing error which will be corrected. Panel a) should read "Loss of volume above flotation (Gt)" and panel b) should read "Change in total ice mass (Gt)". This entire section will be expanded to include results from the other models, and to improve clarity.

• Line 460: I agree with the authors' choice of capping the extreme values when calculating the correlation. Is this something you did for all the correlation values you got? I think it could be worth capping extremely low and high values. From my experience, $10^6$ Pa m$^{-1/3}$ a$^{1/3}$ is a good value for capping low friction coefficient but the threshold values below and above which the flow is virtually not affected could be tested in a more systematic way to see if it could increase the Pearson correlation coefficients.

Several thresholds were tried for capping extreme values, and the chosen values are those which were deemed to strike the best balance between preserving the shape of the data and discarding anomalous spikes. This was not done for every correlation value, but only where necessary in some of the comparisons presented in Table A1. The affected values are indicated in the table and caption.

**References**

Brondex, J., Gillet-Chaulet, F. and Gagliardini, O., 2019. Sensitivity of centennial mass loss projections of the Amundsen basin to the friction law. *The Cryosphere*, *13*(1), pp.177-195.

Morlighem, M., Seroussi, H., Larour, E. and Rignot, E., 2013. Inversion of basal friction in Antarctica using exact and incomplete adjoints of a higher-order model. *Journal of Geophysical Research: Earth Surface*, *118*(3), pp.1746-1753.

Goldberg, D., Narayanan, S.H.K., Hascoet, L. and Utke, J., 2016. An optimized treatment for algorithmic differentiation of an important glaciological fixed-point problem.

Goldberg, D.N., Gourmelen, N., Kimura, S., Millan, R. and Snow, K., 2019. How accurately should we model ice shelf melt rates?. *Geophysical Research Letters*, *46*(1), pp.189-199.

---

## Author Response (AR2)

**The transferability of adjoint inversion products between different ice flow models**
**Minor revisions - response to referee comments**

Author responses written in blue.

In general, we have made small adjustments to the text where required by the referees, and have updated Fig.6 and Fig.7 according to suggestions.
We have split section 5.1 into two parts, results and discussion, as it was felt a clearer distinction was needed.
We have also added well-deserved acknowledgements to the two reviewers and the editor for their efforts to help us improve the quality of our paper!

**Referee #1**

This paper has improved significantly since my first review.
My comments are now mostly minor in nature.

Line 110. Completely optional, but it would be simpler to have a single equation (e.g., equation 2), with generic values for the coefficient (e.g., D) and vo, then simply say in Ua D=(Co+C)^(), in ISSM and Streamice D=beta^2. UA and SI use values of vo of xx and yy, respectively. ISSM uses vo=0 but….

We have decided to keep the original formatting, for a simple visual comparison of the sliding laws.

Line 194. Not clear to me what it means "hydrostatically inverting for the bed" (is this on grounded or floating ice). If on grounded, please elaborate more. If just on the shelf, please say inverting for thickness (or the bottom of or thickness of the ice shelf – not the bed, which implies rock, not water).

This sentence has been rewritten for clarity.

Line 212 to better justify the assumption would be good to add …parts of Ua's BC to zero, WHICH GENERALLY FOLLOWS THE CATCHMENT DIVIDE (assuming this is the case).

This is a good suggestion, and has been added to the text.

Line 230 – the part about interpolating to streamice. This seems true for some figures, but not for others. Why not just use the instersection of the two domains when comparing (as in earlier figures).

In all figures, the largest amount of non-extrapolated data as possible is included in the plots. The mask based intersection of domains is only applied to pairwise comparisons of differences between two models. The text has been updated to clarify this choice.

Line 348. The evolution… This sentence reaches a conclusion based on the results, before identifying what results are being discussed. Move sentence from line 351 before (i.e., so Figure 6 is called out first).

The sentence introducing Fig.6 has been moved to the start of the paragraph.

Figure 6. Please show each color line style in the legend, which should fit (use a two-column legend if you have to). It's too confusing the way it's currently presented.

The legend on Fig.6 has been updated as requested.

Figure 7. Please put "Forward …" along the lefthand edge, rather than right. Consider using separate color tables for floating and grounded ice. As shown, it says more about the melt conditions than the model response.

Fig.7 has been updated with a different colour scale to make changes over grounded ice more visible, and to move the row titles to the left side.

General comment about the last couple of pages. I found the discussion of the comparisons with other MIPs to be to apples to oranges. You have far fewer models and less variation in parameters. That doesn't mean your findings aren't significant; just you are reaching a bit too far to provide context. Also at the top of page 17, the ISMIP6 results cited here cover 85, not 100 years. They also cover a huge range of melt and SMB forcing. If you look at the results in that paper for fixed forcings, applied to WAIS, the range is much smaller. Please cite the more relevant numbers (see Figure 8-11).

There are certainly limitations to these types of comparison, which we have tried to make clear in the text. Regarding the initMIP paper we cite, the results are over 100 years (perhaps there is another paper as part of ISMIP6 which contains 85 year experiments?). We refer only to the control experiment in that paper, in which climate forcing was kept constant. The text has been updated to make this more clear. Explicit values for the more relevant comparison (the two ISSM runs included in initMIP) have been added to the text as suggested.

Referee #2

Thank you to the authors for the work on this revision and their great attention to details. The authors have clearly accounted for my previous comments, running a significant amount of new simulations and clearly detailing modelling choices and inherent limitations of each model. I think this extra work has strengthened the manuscript and gives the reader more transparency on what to expect when transferring inversion products from one model to another. I have only a few additional comments, which are suggestions rather than critical remarks.

In the end, I am very happy with this version of the manuscript. Like the authors, I particularly think that the results of the forward modeling over a 40-year period exhibiting ~10 to ~30-40% difference in SLR contribution is very encouraging for the application of inversion product transfer in the future. As underlined by the authors, I am sure that this work will be useful to many other modelers interested in such transfer of inversion product, either to save computing time (by reusing outputs from previous studies) or for intercomparison experiments.

[Abstract:] If space is not a problem, I would also add the value in mm in addition to the percentage variability of sea level.
• Line 24: specify the location of Ice Stream E (e.g., tributary of Ross Ice Shelf, Antarctica).
• Line 29-32: add a reference to Gillet-Chaulet (2020) who shows a promising ensemble Kalman filter method.
• Line 74: I would specify the period of observation: "[…] and has been accelerating over the last decades (Sutterley et al., 2014)"
• Line 75: I would change "can produce" for "produce"

All of the above suggested changes have been made to the text.

• Line 92 to 97:
o Although the paper is really aimed at ice-sheet modelers that know the different iceflow equations, I would give a small explanation about what is L1L2. Maybe one sentence stating the hypothesis and differences between SSA and L1L2.
o I would also specify here that ISSM and Ua are both FE models (since you introduce STREAMICE as "not a purely FE model" in line 157). Also indicate why STREAMICE is not a purely FE model.

Additional detail has been added to explain this description of STREAMICE, and the L1L2 approximation.

• Line 105: I think "rate factors are […]" should be singular
• Line 179: "inversion processes have performed" instead of "[…] has performed"
• Line 226: change "inversions run" run for "inversion runs" or just "inversions" (same in the caption of Figure B2)

All of the above suggested changes have been made in the text.

• Line 246-248 and Appendix B: It is interesting to see that Ua misfit largely increases when using B from ISSM and only inverts for $\beta_2$ (Figure 3b and B2c look really alike, especially when looking at the misfit on the Thwaites glacier tongue). I think you rightly point out that, in addition to the logarithmic cost function used in ISSM, the fact that the inversion of both parameters is sequent (i.e., B and then $\beta_2$) and not simultaneous could have a large impact.
• Line 294: I think that this statement could be linked with my previous comment: $\beta_2$ is fairly similar in all the models but the inversion of B in ISSM seems to suffer from the fact that $\beta_2$ is only using its prior value when inverting for B. Did you try to invert

for $\beta_2$ first and then B, or even reinverting for B a second time (I detail this a bit further)? Notice that I understand STREAMICE and ISSM showing similar B values on the shelve maybe suggest that this is not that important.

Inverting the sequence in a different order or reinverting for B were not attempted. This is certainly an interesting idea, and one which the ISSM users among us are interested to try in future. However, we have not added any new experiments to this manuscript.

• Line 326: Should not it be "sequential nature of ISSM" instead of "non-sequential" (since ISSM uses a sequence of B and $\beta_2$ inversion)?

This has been corrected.

• Line 375: what do you expect or see as differences between the GL retreat of SSA models and the L1L2? I think that this should be slightly develop since you point out the difference in the equations.

This sentence was actually in the wrong place in the paragraph, as it was supposed to be a separate point from the GL retreat. The paragraph has been rearranged.

• Line 424: "They each use different methods and employ different techniques during the inversion process, […]" looks a bit repetitive, maybe change for "They each use different inversion techniques […]"?
• Line 463: add a comma between "[…] are selected" and "the corresponding […]"
• Line 492-493: please consider adding the mean magnitude misfit obtained with the Interior Point algorithm (for direct comparison with the value you give for M1QN3).
• Line 516: "to be" instead of "be be"

All of the above suggested changes have been made to the text.

• Line 519: remind here the value of the Pearson correlation so that the reader can directly make the comparison with the value 0.513 given in line 522.

The Pearson coefficients (including those between the original experiments) are all displayed in Table B1, which is referred to at the end of the paragraph.

• Figure B3: Why is ISSM $\beta_2$ so low over Pine Island fast ice when using Priors 1? Maybe add a potential explanation?

A potential explanation has been added, attributing the change to the uniform values of B in Priors1. Since Úa is the only model which inverts for the values of B over the grounded ice, it is the model least affected by the choice of priors. In fact, it may well be the case that with a good choice of B, the prior for beta^2 would not have much affect on the outputs from ISSM or STREAMICE. However, we do not have the necessary results to prove such a hypothesis.

As it is highlighted in the paper, the "bad" result of the steady misfit in ISSM is not necessarily a problem when looking at the evolution of the misfit after a forward simulation. Maybe it is worth to clearly state this (maybe in the discussion) ; it is somewhat understood when reading the paper but emphasizing this could be useful.

An extra small paragraph has been added to the discussion to emphasise this point.

The figure 7 in the new version is indeed much more legible. Please, indicate that the black line is the grounding line. Could you also plot the initial position of the grounding line (right after inversion)? It would allow a better visualization of the grounding line migration.

The initial grounding line has been added in Fig.7, and the caption updated to refer to it.

---

## Author Response (AR3)

**The transferability of adjoint inversion products between different ice flow models**
**Minor revisions - response to editor comments**

Author responses written in blue.

Dear Jowan,

Thanks for this revised version of your manuscript which has greatly improved since its nominal submission. I have nevertheless still few comments that should be accounted for before it is accepted for publication in The Cryosphere. They are listed below.

Best regards,
Olivier Gagliardini

Thank you for the comments. Revisions have been made as noted below.

- I think the paper would be easier to follow if sub-sections 3.1 and 3.2 were switched?
These have now been switched.

- line 198: "using the result". Which result? Missing something here?
Updated to clarify that the hydrostatic inversion is using the smoothed surface.

- line 216: it is not clear from this sentence if only the normal velocity or all components are set to zero for Ua on the lateral boundary of the grounded parts (the classical BC on the edges of a drainage bassin would be normal velocity set to zero, i.e. no flux BC).
Changed wording to "all velocities" to make this clearer (both u and v are set to zero in this case).

- line 235: "some areas" -> "these areas" ?
Changed this phrase as suggested.

- line 344: this change in the inversion procedure for STREAMICE should be justified. The way it is presented, it looks suspicious for the reader and breaks the well designed sets of experiments.
The justification has now been added to this section. This was certainly needed! (It was previously mentioned in Appendix B4, but had not been referred to in the main text where it should have been)

- line 352 and at many other places in the manuscript: "in Figure 6" -> "in Fig. 6". See https://www.the-cryosphere.net/submission.html#figurestables (The abbreviation "Fig." should be used when it appears in running text and should be followed by a number unless it comes at the beginning of a sentence, e.g.: "The results are depicted in Fig. 5. Figure 9 reveals that...".
The references have been changed to comply with submission guidelines. All

occurrences of Figure, Section and Equation have been updated to Fig., Sect. and Eq., except at the start of a sentence. Consistent capitalisation of geographic terms in names (eg. Thwaites Glacier) has also been applied throughout, including updating Fig.1.

- line 366: How the friction below ice-shelves is treated in each model in the case of a GL advance should be explained in the model description
Úa's different treatment of friction is now addressed at the start of section 5, along with the difference in STREAMICE priors. This seems the most appropriate place to introduce it, as it was a choice made for these experiments, rather than being a general feature of the model. The original paragraph in Sect. 5.1 has been altered to reflect this.